# StructMAR: Structure-Aware Masked Autoregression for Explicit Layout Alignment in Text-to-Image Generation

**Gang Cao** [1]   **Junying Zhang** [1]

## Abstract

Although text-to-image generation has achieved significant progress, strict instance-level layout alignment remains a challenge for many applications. Masked Autoregressive (MAR) models on continuous latents are both efficient and high-fidelity, yet the standard practice of flattening 2D latents into 1D sequences weakens spatial topology, limiting precise controllability. To address this, we propose StructMAR, a structure-aware masked autoregressive framework that transforms layout alignment from a soft correlation into an explicit structural alignment. By integrating 2D Rotary Positional Embeddings with a Layout-Guided Attention Bias, StructMAR explicitly biases latent tokens toward their corresponding layout instances during attention computation. We further use Group Relative Policy Optimization (GRPO) as a final-stage policy refinement to reduce the mismatch between the MAR training objective and detector-based layout evaluation metrics. Evaluated on the COCO-Position and COCO-MIG benchmarks, StructMAR achieves state-of-the-art performance, reaching 57.2 AP and 79.4 mIoU on the former, and 61.7 ISR and 56.9 mIoU on the latter. These results, coupled with a $4.05\times$ inference speedup, underscore the efficacy of explicit structural inductive biases in controllable autoregressive generation.

## 1. Introduction

Text-to-image generation has advanced rapidly and is increasingly used in practical content creation workflows (Rombach et al., 2022; Ramesh et al., 2022). In applications such as UI design, poster layout, and multi-object scene synthesis, the user's requirement often goes beyond producing a plausible image: the generated content is expected to follow explicit geometric instructions with high fidelity. A compact and widely adopted interface for such control is instance-level layouts specified by bounding boxes, where each box indicates both the target category and the spatial region that should be respected (Sun & Wu, 2019). In this setting, a central distinction emerges between *layout conditioning*—providing layout information as context—and *layout alignment*—ensuring the generation consistently obeys the layout constraints. In practice, many failures arise when bounding boxes are treated as soft hints rather than executable rules, especially for dense, overlapping, or small-object layouts where spatial ambiguities are amplified.

Existing layout-controlled generation methods have been dominated by diffusion-based approaches (Zhang et al., 2023), including strong baselines that incorporate region-level conditions and attention guidance (Li et al., 2023; Xie et al., 2023). These methods can produce high-quality images and achieve competitive alignment in many cases, but their inference typically involves multiple denoising steps, and the adherence to layout constraints is often mediated by implicit attention dynamics rather than explicit enforcement. In parallel, Masked Autoregressive models (MAR) over continuous latents have emerged as an efficient alternative backbone that combines parallel masked prediction with high visual fidelity (Chang et al., 2022; Li et al., 2024). However, MAR models commonly flatten 2D latent grids into 1D token sequences and apply generic transformer attention. This design can weaken spatial topology and makes it difficult to maintain strict instance-level correspondence throughout the network. Consequently, even when layouts are injected via concatenation or generic cross-attention, the intended correspondence may be diluted across layers or overridden by stronger semantic signals. Importantly, this suggests that the main limitation is not the availability of layout signals, but the lack of an explicit attention-level mechanism in MAR that biases token-to-instance layout correspondence during attention computation.

Motivated by this gap, we argue that enabling strict layout alignment in MAR-style masked generation requires

---

[1]School of Computer Science and Technology, Xidian University, Xi'an, Shaanxi, China. Correspondence to: Junying Zhang <jyzhang@mail.xidian.edu.cn>.

*Proceedings of the 43rd International Conference on Machine Learning*, Seoul, South Korea. PMLR 306, 2026. Copyright 2026 by the author(s).

explicit box-compatible biasing within transformer attention logits, rather than relying on soft conditioning to learn correspondence implicitly. Concretely, we propose **Struct-MAR**, which turns layout alignment from an implicitly learned correlation into an **explicit latent-to-layout structural execution mechanism** within transformer attention. This perspective shifts the role of layout inputs from optional context to an actionable constraint that is executed during generation, making adherence more stable under challenging layouts.

StructMAR integrates three complementary components to support this goal. First, we introduce 2D rotary positional embeddings (2D RoPE) (Heo et al., 2024) for continuous latent tokens to recover 2D relative spatial awareness that is obscured by sequence flattening. This component primarily addresses the *where* question—representing spatial relations reliably across the latent grid. Second, and most critically, we design a **Layout-Guided Attention Bias** implemented as block-wise logit injection: bounding-box membership is translated into attention-logit biases so that latent tokens are encouraged to attend in a manner consistent with the instance-level layout. This provides direct structural guidance, turning layout alignment into an explicit part of attention computation rather than a purely soft condition. Third, to avoid degrading semantic coherence when structural constraints are strong, we add a *semantic safeguard* via gated cross-attention, treating text semantics as controlled context while preventing it from dominating the structural execution pathway.

While the proposed attention-level structural bias already provides a strong basis for layout adherence, the supervised MAR objective remains reconstruction-oriented and does not directly optimize evaluation-time metrics such as AP and mIoU. We therefore use **Group Relative Policy Optimization (GRPO)** (Shao et al., 2024) as a final-stage policy refinement, rather than as the primary source of the architectural contribution. Specifically, we treat masked latent recovery as a stochastic generation trajectory and optimize detector-based geometric rewards. This helps refine boundary-level layout decisions while preserving the core generation mechanism of StructMAR.

We evaluate StructMAR on two complementary benchmarks that emphasize layout fidelity, robustness in crowded scenes, and inference efficiency. On COCO-Position (Zhou et al., 2024), StructMAR achieves **57.20 AP** and **79.40 mIoU**, surpassing strong diffusion baselines such as MIGC (54.69 AP, 77.38 mIoU) while maintaining competitive image quality (FID-6K 24.70 vs. MIGC's 24.52).

On COCO-MIG (Zhou et al., 2024), which varies the number of instances per image from L2 to L6, StructMAR demonstrates superior robustness as layouts become increasingly dense. While performance is comparable to

MIGC in simpler cases (L2), StructMAR exhibits significantly stronger stability than diffusion-based methods under crowded settings; for instance, it achieves an L6 success rate of **62.80** compared to 56.88 for MIGC. This leads to higher overall average success rates and mIoU. Finally, StructMAR is substantially faster during inference on this benchmark (**3.85s** vs. 15.61s for MIGC, representing a 4.05× speedup). These results suggest that masked autoregressive decoding over continuous latents is a highly promising direction for efficient and controllable generation.

Our contributions are summarized as follows:

- **Gap identification in MAR for strict alignment.** We highlight that flattened MAR backbones with generic attention lack an explicit attention-level mechanism for layout-aware token-to-instance correspondence, making soft conditioning unreliable under dense and overlapping layouts.

- **Latent-token-to-instance correspondence via attention-logit injection.** We implement a Layout-Guided Attention Bias through block-wise logit injection, which explicitly strengthens latent-to-layout correspondence within the $\mathbf{X} \to \mathbf{L}$ attention block while preserving latent-to-latent interactions for global coherence.

- **Structure–semantics coordination.** We incorporate 2D RoPE as a necessary spatial enabler and a gated semantic pathway to maintain semantic coherence while executing structural constraints.

- **Metric-aligned fine-tuning for layout fidelity.** We apply GRPO with detector-based rewards to align training with evaluation-time layout metrics, improving alignment and helping rebalance quality in the final model.

## 2. Related Work

We study layout-controlled (bbox-conditioned) text-to-image generation with an emphasis on layout alignment, i.e., ensuring generated pixels comply with instance-level geometric constraints. Our focus is on enforcing pixel-to-layout consistency within transformer attention, rather than relying on shallow concatenation of layout tokens.

**Layout-conditioned text-to-image generation.** Controllable text-to-image diffusion methods can be categorized by where and how control signals are injected (e.g., input/feature/attention/denoising-time) (Cao et al., 2025; Jiang et al., 2024). For instance-level layout control, existing methods either add layout-conditioning branches or adapters to pretrained generators, as in ControlNet, or inject region-/box-level signals through gated attention modules, as in GLIGEN (Li et al., 2023). Inference-time methods can

also enforce box constraints without finetuning (e.g., BoxDiff) (Xie et al., 2023). Despite these advances, many methods still treat layout as a soft condition (feature injection or generic cross-attention), leaving the coupling between pixel tokens and instance constraints under-constrained, which can degrade adherence under dense, overlapping, or small-object layouts.

**Recent layout-aware and autoregressive baselines.** Recent layout-to-image methods further improve layout and compositional control, especially under complex or overlapping layouts. However, their reported protocols are not always directly comparable with the full COCO-Position and COCO-MIG protocols used by MIGC and followed in our main experiments. We therefore focus the main comparisons on strictly matched COCO protocols. In parallel, recent continuous-token autoregressive backbones such as SphereAR (Ke & Xue, 2025) improve the latent representation of autoregressive image generation. Our focus is complementary: rather than replacing the MAR backbone, we isolate how explicit latent-to-layout structural execution affects layout-conditioned generation under the same backbone.

**Masked / autoregressive image generation backbones.** Masked token modeling has re-emerged as an efficient paradigm for image generation. MaskGIT predicts randomly masked visual tokens with iterative parallel decoding (Chang et al., 2022), and Muse scales masked generative transformers for text-to-image generation (Chang et al., 2023). Moving beyond discrete VQ tokens, MAR models per-token likelihoods in a continuous space via a diffusion-style token loss (Li et al., 2024). Recent work strengthens MAR-style masked generation with bidirectional backbones and 2D RoPE (e.g., MaskGIL) (Xin et al., 2025). While these backbones provide strong generative priors and scalability, they are largely structure-agnostic: naive layout conditioning can be diluted across depth, limiting strict pixel-to-box alignment.

**Structural inductive bias: 2D spatial encoding and attention-level constraints.** Accurate layout adherence requires both representing 2D spatial relationships and enforcing layout constraints during token interactions. For spatial encoding, flattening images into 1D sequences obscures 2D topology; RoPE has been analyzed for vision and 2D grids and shows practical benefits when implemented for 2D visual tokens (Heo et al., 2024), and recent masked generators adopt 2D RoPE for improved spatial reasoning (Xin et al., 2025). For constraint enforcement, most layout-to-image methods inject layout as tokens/features (including gated region-level conditioning (Li et al., 2023)) or rely on generic cross-attention, which still leaves the model free to ignore instance geometry. A more direct route is to impose explicit pixel-to-box compatibility on attention logits (e.g., box-aware bias/masking), so tokens inside a box are encouraged or forced to attend to the corresponding instance representation (Xiang et al., 2025). We follow this line by coupling 2D RoPE with layout-guided attention bias to encode pixel-to-layout constraints directly in transformer attention.

**Optimizing layout fidelity with reinforcement learning (GRPO).** Supervised training may not directly optimize evaluation-time layout metrics such as detection IoU or centroid alignment. RL-style finetuning has been used to optimize non-differentiable objectives in generative models (e.g., DDPO for diffusion) (Black et al., 2023), and preference/group-based objectives improve efficiency and stability (Wallace et al., 2024; Luo et al., 2025). Critic-free updates such as GRPO provide scalable alternatives to PPO-style critic training (Shao et al., 2024). In our setting, we view masked generation as a stochastic policy over image-token trajectories and employ GRPO with detector-based rewards to directly optimize layout fidelity, complementing attention-logit constraints.

In summary, while prior controllable generation methods often rely on soft layout conditioning, we target strict layout alignment by integrating 2D spatial encoding with attention-level geometric constraints inside a masked autoregressive transformer, and further improve metric alignment using GRPO-based reward optimization.

## 3. Method

### 3.1. Problem Setup and Notation

We study layout-controlled text-to-image generation in a continuous latent setting. The input consists of a text prompt and instance-level layouts specified by bounding boxes (e.g., defining a "cat" in the top-left region). Our goal goes beyond treating layouts as soft conditions: we target *layout alignment*, i.e., latent tokens at spatial locations must remain aligned with their owning instance constraints throughout transformer computation.

**Text tokens.** Let the text prompt be a token sequence

$$\mathbf{S} = \{\mathbf{s}_k\}_{k=1}^M. \tag{1}$$

**Instance layouts.** Layout conditions are given by

$$\mathcal{B} = \{(b_j, c_j)\}_{j=1}^N, \tag{2}$$

where each bounding box is denoted by $b_j = (x_1, y_1, x_2, y_2) \in [0, 1]^4$ with $x_1 < x_2$ and $y_1 < y_2$, specifying the normalized top-left and bottom-right corners, respectively.

**Continuous latent tokens.** We use a pre-trained VAE to map an image into a continuous latent grid

$$\mathbf{Z} \in \mathbb{R}^{H \times W \times d}, \tag{3}$$

which is flattened into a sequence of continuous image latent tokens

$$\mathbf{X} = \{\mathbf{x}_i\}_{i=1}^{HW}, \quad \mathbf{x}_i \in \mathbb{R}^d. \tag{4}$$

Each token $\mathbf{x}_i$ corresponds to a spatial coordinate $(u_i, v_i)$ (grid indices or normalized coordinates).

**Layout tokens.** Instance layouts are encoded into embeddings

$$\mathbf{L} = \{\boldsymbol{\ell}_j\}_{j=1}^{N}, \quad \boldsymbol{\ell}_j = f_{\text{layout}}(b_j, c_j). \tag{5}$$

*Take-away.* We generate continuous image latent tokens conditioned on text and instance layouts, enforcing latent-token-to-layout alignment strictly within transformer attention.

## 3.2. Preliminaries: Diffusion-based Masked Autoregression (MAR)

We build upon Masked Autoregressive Modeling (MAR) for continuous latents. MAR randomly masks a subset of latent tokens and learns to recover them via a diffusion-style denoising objective.

Let $m \subset \{1, \dots, HW\}$ be masked indices, $t$ a diffusion timestep, $\mathbf{X}_m^{(t)}$ the noisy version of masked tokens, and $\mathbf{X}_{\neg m}$ the observed tokens. Let $\mathbf{c} = (\mathbf{S}, \mathcal{B})$ denote the conditioning information. We use the standard abstract MAR loss:

$$\mathcal{L}_{\text{MAR}} = \mathbb{E}_{\mathbf{X},m,t,\epsilon} \left[ \sum_{i \in m} \left\| \epsilon - \epsilon_\theta(\mathbf{X}_m^{(t)}, \mathbf{X}_{\neg m}, \mathbf{c})_i \right\|_2^2 \right], \tag{6}$$

where $\epsilon$ is the injected noise and $\epsilon_\theta(\cdot)_i$ predicts the noise (or an equivalent denoised latent) for token $i$.

*Limitation.* While MAR provides a strong generative prior, it typically relies on 1D positional encodings and treats condition $\mathbf{c}$ as generic context. This renders the backbone structure-agnostic, providing no explicit inductive bias for strict layout alignment.

## 3.3. StructMAR Architecture

To address the lack of spatial topology and explicit structural constraints, we propose StructMAR, a structure-aware masked autoregressive architecture.

Our transformer operates on the joint sequence $[\mathbf{L}; \mathbf{X}]$ to enable direct interaction between layout and image tokens. We introduce explicit structural inductive biases (illustrated as the *Structural Priors* module in Fig. 1) at two levels: the representation level via 2D RoPE, and the interaction

level via Layout-Guided Attention Bias, while injecting text semantics via gated cross-attention. The detailed structure of our Transformer block is depicted in Figure 2.

**Different roles of layout and text tokens.** Layout and text conditions are deliberately injected through different pathways. Layout tokens $\mathbf{L}$ are concatenated with image latent tokens $\mathbf{X}$ and participate directly in structure-aware self-attention over $[\mathbf{L}; \mathbf{X}]$, making them part of the executable geometric pathway. In contrast, text tokens $\mathbf{S}$ are injected through a gated cross-attention branch after structure-aware attention, providing semantic guidance without overriding the latent-to-layout structural pathway. This separation is important because layout tokens define where instance-level constraints should be executed, whereas text tokens provide what semantic content should be generated.

### 3.3.1. 2D ROTARY POSITIONAL EMBEDDINGS (2D RoPE)

Standard flattening of the $H \times W$ latent grid into a 1D sequence obscures intrinsic 2D topology. We replace learnable absolute embeddings with 2D Rotary Positional Embeddings (RoPE 2D) to inject geometric awareness.

Unlike additive embeddings, RoPE encodes position by rotating the query and key vectors in the complex embedding space. For a latent token $\mathbf{x}_i$ at 2D coordinate $(u_i, v_i)$, we split its head dimension $d$ into two halves, denoted as $\mathbf{Q}_i^{(1)}$ and $\mathbf{Q}_i^{(2)}$. We apply a rotation transformation parameterized by the column index $u_i$ to the first half, and by the row index $v_i$ to the second half:

$$\begin{aligned}
\tilde{\mathbf{Q}}_i &= \text{cat}\left(\mathbf{Q}_i^{(1)}\mathbf{R}(u_i), \ \mathbf{Q}_i^{(2)}\mathbf{R}(v_i)\right), \\
\tilde{\mathbf{K}}_i &= \text{cat}\left(\mathbf{K}_i^{(1)}\mathbf{R}(u_i), \ \mathbf{K}_i^{(2)}\mathbf{R}(v_i)\right),
\end{aligned} \tag{7}$$

where $\mathbf{R}(\cdot)$ is a unitary rotation matrix. This formulation makes the positional phase in the query–key interaction depend on the relative 2D offsets $(u_i - u_j)$ and $(v_i - v_j)$, rather than only on absolute positions. While 2D RoPE provides useful spatial awareness, it remains a positional bias and is insufficient for layout adherence on its own.

### 3.3.2. GATED CROSS-ATTENTION FOR SEMANTICS

Naively concatenating text and layout tokens risks entangling semantic control with geometric constraints. We inject textual semantics $\mathbf{S}$ via gated cross-attention after structure-aware attention:

$$\mathbf{h} \leftarrow \mathbf{h} + g \cdot \text{CrossAttn}(\mathbf{h}, \mathbf{S}), \qquad g = \tanh(\gamma), \tag{8}$$

where $\mathbf{h}$ denotes image-latent hidden states and $\gamma$ is initialized to zero (thus $g \approx 0$ at the beginning of training).

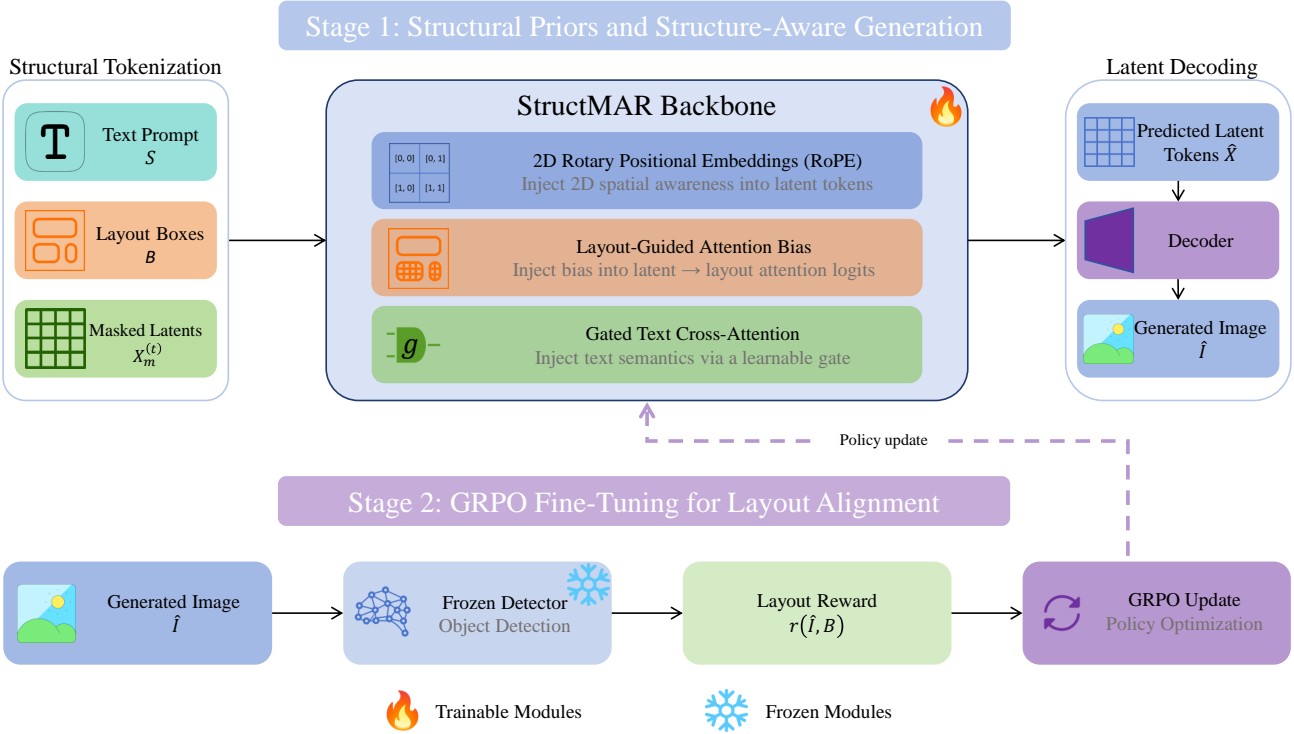

*Figure 1.* **Overall Architecture of StructMAR.** Our framework integrates structural inductive biases into a Masked Autoregressive (MAR) pipeline. It consists of three primary stages: (1) **Structural Tokenization**, where 2D RoPE and Layout Tokens provide explicit spatial grounding; (2) **Structure-Aware Generation**, where the transformer executes layout-guided constraints via attention-logit injection; and (3) **Metric-Aligned Optimization**, utilizing Group Relative Policy Optimization (GRPO) with detector-based rewards (IoU and confidence) to directly optimize layout fidelity without a critic network.

This "semantic safeguard" allows semantics to act as controllable context without dominating the structure-aware layout pathway.

### 3.4. Explicit Layout Alignment via Layout-Guided Attention Bias

While 2D RoPE provides spatial awareness, it does not by itself ensure layout adherence. We introduce Layout-Guided Attention Bias, an explicit box-compatibility bias injected directly into the layout-attention logits.

**Layout bias construction.** We append a background layout token $\ell_0$ for latent tokens outside all boxes and use $\bar{\mathbf{L}} = [\ell_0; \mathbf{L}]$. We construct $\mathbf{M}^{\text{layout}} \in \{0, -\infty\}^{HW \times (N+1)}$ to encode latent–layout compatibility: a latent token at $(u_i, v_i)$ receives bias 0 for layout tokens whose boxes contain it, or for $\ell_0$ if it is outside all boxes, and receives $-\infty$ for incompatible tokens. In practice, $-\infty$ is implemented as a large negative constant before softmax.

**Block-wise logit injection.** Attention is computed on the joint sequence $[\bar{\mathbf{L}}; \mathbf{X}]$, where $\bar{\mathbf{L}} = [\ell_0; \mathbf{L}]$ includes the background layout token. For notational simplicity, we still denote the layout-token block by $\mathbf{L}$ in the attention blocks

below. Let

$$\mathbf{A} = \frac{\tilde{\mathbf{Q}}\tilde{\mathbf{K}}^\top}{\sqrt{d}} \tag{9}$$

be the RoPE-modified attention logits, partitioned into blocks:

$$\mathbf{A} = \begin{bmatrix} \mathbf{A}_{LL} & \mathbf{A}_{LX} \\ \mathbf{A}_{XL} & \mathbf{A}_{XX} \end{bmatrix}. \tag{10}$$

We inject the layout bias only into the $\mathbf{X} \to \mathbf{L}$ logit block:

$$\mathbf{A}' = \begin{bmatrix} \mathbf{A}_{LL} & \mathbf{A}_{LX} \\ \mathbf{A}_{XL} + \mathbf{M}^{\text{layout}} & \mathbf{A}_{XX} \end{bmatrix}, \tag{11}$$
$$\text{Attn} = \text{Softmax}(\mathbf{A}')\mathbf{V}.$$

*Key insight.* We constrain latent-token-to-layout interactions (the bipartite edges $\mathbf{X} \to \mathbf{L}$), rather than masking latent-to-latent self-attention ($\mathbf{X} \leftrightarrow \mathbf{X}$). This restricts the layout-token candidates available to each latent token within the $\mathbf{X} \to \mathbf{L}$ block, encouraging spatially compatible latent-to-instance interactions while preserving unrestricted latent-to-latent interactions for global coherence.

*Complexity.* Constructing $\mathbf{M}^{\text{layout}}$ costs $O(HW \cdot N)$, but typically $N \ll HW$; with vectorized box-membership tests, the overhead is negligible in practice.

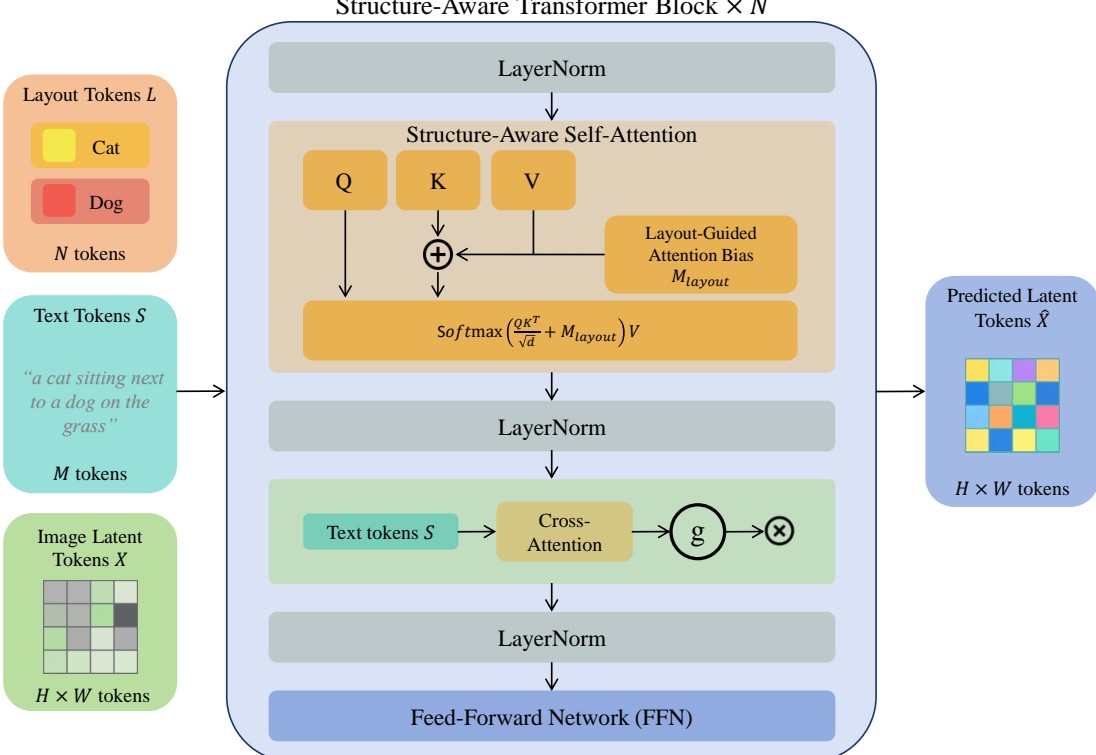

*Figure 2.* **Structure-aware Transformer block.** Layout tokens and image latent tokens are processed in the structural pathway, where the proposed $\mathbf{X} \rightarrow \mathbf{L}$ layout-guided attention bias explicitly strengthens token-to-instance correspondence. Text tokens are injected through a separate gated cross-attention branch, providing semantic guidance without overriding the structural pathway.

### 3.5. Optimization via Group Relative Policy Optimization (GRPO)

The supervised MAR objective does not directly optimize discrete layout metrics such as IoU. We therefore fine-tune the model using reinforcement learning.

**Policy view.** We treat the masked denoising generation of continuous latents as a stochastic policy $\pi_\theta$ over token trajectories conditioned on $(\mathbf{S}, \mathcal{B})$. A trajectory corresponds to the sequence of iterative masked denoising steps until all tokens are recovered and decoded.

**Dense layout reward.** We design a reward focusing on geometric fidelity:

$$R = \lambda_{\text{IoU}} \cdot \text{IoU}_{\text{det}} + \lambda_{\text{conf}} \cdot \text{Conf}_{\text{det}} - \lambda_{\text{cent}} \cdot \text{Dist}_{\text{centroid}}, \quad (12)$$

where $\text{IoU}_{\text{det}}$ and $\text{Conf}_{\text{det}}$ are computed using a frozen detector on the generated image, and $\text{Dist}_{\text{centroid}}$ penalizes centroid offsets.

**Reward detector and evaluator separation.** Unless otherwise specified, we use Faster R-CNN as the reward detector during GRPO and keep GroundingDINO as the fixed evaluation detector following the MIGC protocol. This separation reduces the risk of directly optimizing for the

same detector used at evaluation time. We further evaluate alternative reward detectors, including YOLOv8 and GroundingDINO, in Sec. 4.4.

**Critic-free optimization.** For each condition $(\mathbf{S}, \mathcal{B})$, we sample a group of $G$ masked decoding trajectories $\{\tau_g\}_{g=1}^{G}$ and compute detector-based rewards $\{R_g\}_{g=1}^{G}$. Rewards are normalized within the group to obtain relative advantages $A_g$, avoiding an explicit critic. The GRPO update is approximated as

$$\nabla_\theta \mathcal{J}(\theta) \approx \sum_{g=1}^{G} A_g \nabla_\theta \log \pi_\theta(\tau_g | \mathbf{S}, \mathcal{B}). \quad (13)$$

For continuous-latent MAR, the trajectory log-probability is accumulated from Gaussian token-update likelihoods over masked decoding steps, with an optional KL penalty to the supervised model. Details are provided in Appendix A.6.

## 4. Experiments

In this section, we strictly follow the evaluation protocols established by prior state-of-the-art methods, specifically GLIGEN (Li et al., 2023) and MIGC (Zhou et al., 2024), to ensure fair comparison and reproducibility. We assess StructMAR on three key dimensions: spatial alignment

precision, robustness in complex multi-instance scenarios, and inference efficiency.

## 4.1. Experimental Setup

### 4.1.1. DATASETS AND TRAINING PROTOCOL

We utilize the COCO 2014 train split (Lin et al., 2014) for training, avoiding potential data leakage or incomparability issues associated with the 2017 split. The conditioning schema consists of global captions, instance labels, and bounding boxes. Our training objective integrates Layout-Guided Attention Bias to inject structural constraints and utilizes Group Relative Policy Optimization (GRPO) (Shao et al., 2024) to align the generation policy with detector-based layout metrics (e.g., IoU).

### 4.1.2. BENCHMARKS

We employ two distinct benchmarks to evaluate different aspects of controllability:

- **COCO-Position** (Zhou et al., 2024): Following the MIGC protocol, this benchmark evaluates the model's ability to precisely align objects with bounding boxes and maintain image quality.

- **COCO-MIG** (Zhou et al., 2024): Also following the MIGC protocol, this benchmark assesses robustness in multi-instance generation across varying levels of complexity (from 2 to 6 instances per image). Each instance includes a color attribute in addition to its category and bounding box, and we encode the colored instance description, e.g., "red cat", as the instance text tag $c_j$.

### 4.1.3. METRICS

- **Spatial Precision:** We report AP (Average Precision), $AP_{50}$, and mIoU (mean Intersection over Union). We employ Grounding-DINO (Liu et al., 2024) as the primary evaluator. Following standard protocols (Zhou et al., 2024), we report these metrics to measure the geometric alignment between generated objects and user-specified bounding boxes.

- **Image Quality & Semantics:** We report FID-6K (Heusel et al., 2017) to assess perceptual fidelity and CLIP Score (Ramesh et al., 2022) to measure semantic consistency between the image and the text caption.

- **Robustness:** On COCO-MIG, we report Instance Success Rate and mIoU categorized by instance count (L2–L6). We follow the MIGC attribute-aware evaluation protocol: an instance is counted as successful only when its category, position, and color attribute are

correct, and its IoU is set to zero if the color attribute is incorrect.

## 4.2. Results on COCO-Position

We first evaluate the closed-set layout alignment performance on the COCO-Position benchmark. Quantitative results are summarized in Table 1.

**Alignment Performance.** As shown in Table 1, StructMAR achieves a new state-of-the-art in spatial precision metrics. Specifically, our method attains an AP of 57.20 and an mIoU of 79.40. Compared to the strong diffusion-based baseline MIGC (54.69 AP, 77.38 mIoU), StructMAR demonstrates a consistent improvement of +2.51 AP and +2.02 mIoU. The gains are even more substantial when compared to GLIGEN (+16.52 AP). These results suggest that the proposed structure-aware mechanisms provide more effective inductive biases for spatial arrangement than the implicit soft-attention mechanisms used in diffusion baselines.

**Image Quality and Semantic Consistency.** While enforcing strict structural constraints often risks degrading image quality, StructMAR maintains competitive fidelity. Our FID-6K is 24.70, which is comparable to MIGC (24.52) and superior to GLIGEN (26.80). Furthermore, the CLIP score remains stable at 24.60 (vs. MIGC's 24.66). This indicates that the substantial improvement in spatial alignment does not compromise the semantic alignment between the generated image and the text prompt.

## 4.3. Robustness and Efficiency on COCO-MIG

To investigate the model's behavior in complex scenarios, we evaluate StructMAR on the COCO-MIG benchmark, categorizing samples by the number of instances (L2 to L6). Results are shown in Table 2.

**Robustness across Complexity Levels.** A detailed analysis of Table 2 reveals a performance crossover between StructMAR and the diffusion-based baseline MIGC. In simpler settings (L2–L3), MIGC achieves a slightly higher Instance Success Rate (e.g., L2: 67.70% vs. 66.15%). We attribute this to the strong generative priors of the diffusion backbone, which excels in sparse layouts without requiring strict structural constraints. However, as complexity increases (L4–L6), StructMAR demonstrates superior robustness. Notably, at the most challenging L6 level, our method outperforms MIGC by a significant margin in Success Rate (62.80% vs. 56.88%, a +5.92% gain). This suggests that the structure-aware design, including 2D RoPE and the $\mathbf{X} \rightarrow \mathbf{L}$ layout bias, helps mitigate spatial confusion in crowded scenes.

**Inference Efficiency.** StructMAR demonstrates a significant advantage in computational efficiency. The average inference time is 3.85s, compared to 15.61s for MIGC. This $4.05\times$ speedup is not due to fewer refinement iterations;

*Table 1.* Quantitative comparison on the COCO-Position benchmark. StructMAR achieves state-of-the-art results on alignment metrics (AP and mIoU), while maintaining competitive image quality. Results for baselines are cited from the MIGC paper (Zhou et al., 2024).

| Method | AP ↑ | AP$_{50}$ ↑ | mIoU ↑ | CLIP ↑ | FID-6K ↓ |
|---|---|---|---|---|---|
| Real Image | 65.97 | 79.11 | 85.49 | 24.22 | - |
| Stable Diffusion (Rombach et al., 2022) | 0.80 | 2.71 | 21.60 | 25.69 | 23.56 |
| TFLCG (Chen et al., 2024) | 1.75 | 6.77 | 28.01 | 25.07 | 24.65 |
| BOX-Diffusion (Xie et al., 2023) | 3.29 | 12.27 | 33.38 | 23.79 | 25.15 |
| Multi Diffusion (Bar-Tal et al., 2023) | 6.72 | 18.65 | 38.82 | 22.10 | 33.20 |
| Layout Diffusion (Zheng et al., 2023) | 23.45 | 48.10 | 57.49 | 18.28 | 25.94 |
| GLIGEN (Li et al., 2023) | 40.68 | 68.26 | 71.61 | 24.61 | 26.80 |
| MIGC (Zhou et al., 2024) | 54.69 | 84.17 | 77.38 | **24.66** | **24.52** |
| **Ours (StructMAR)** | **57.20** | **85.60** | **79.40** | 24.60 | 24.70 |

*Table 2.* Quantitative comparison on the COCO-MIG benchmark. We report Instance Success Rate, mIoU, and inference time. L2–L6 denote different instance-count levels. Avg follows the official COCO-MIG aggregation protocol used in MIGC.

| Method | Instance Success Rate (%) ↑ | | | | | | mIoU ↑ | | | | | | Time (s) ↓ |
|---|---|---|---|---|---|---|---|---|---|---|---|---|---|
| | L2 | L3 | L4 | L5 | L6 | Avg | L2 | L3 | L4 | L5 | L6 | Avg | |
| Stable Diffusion (Rombach et al., 2022) | 6.87 | 5.01 | 3.45 | 3.27 | 2.21 | 3.61 | 18.92 | 17.44 | 15.85 | 15.17 | 14.42 | 15.80 | 9.18 |
| TFLCG (Chen et al., 2024) | 20.47 | 12.71 | 8.36 | 6.72 | 4.36 | 8.62 | 29.34 | 25.06 | 20.82 | 18.81 | 17.86 | 20.92 | 19.92 |
| Box-Diffusion (Xie et al., 2023) | 24.61 | 19.22 | 14.20 | 11.92 | 9.31 | 13.96 | 32.64 | 29.88 | 25.39 | 23.81 | 21.19 | 25.14 | 44.17 |
| Multi Diffusion (Bar-Tal et al., 2023) | 24.88 | 22.14 | 19.88 | 18.97 | 18.60 | 20.12 | 29.41 | 28.06 | 25.59 | 24.83 | 24.71 | 25.89 | 25.15 |
| GLIGEN (Li et al., 2023) | 42.30 | 35.55 | 32.66 | 28.18 | 30.84 | 32.39 | 37.58 | 32.34 | 29.95 | 26.60 | 27.70 | 32.25 | 22.00 |
| MIGC (Zhou et al., 2024) | **67.70** | **59.61** | 58.09 | 56.16 | 56.88 | 58.43 | 59.39 | 52.73 | 51.45 | 49.52 | 49.89 | 51.48 | 15.61 |
| **Ours (StructMAR)** | 66.15 | 58.90 | **60.45** | **61.20** | **62.80** | **61.70** | **61.50** | **56.80** | **57.10** | **55.40** | **56.25** | **56.90** | **3.85** |

*Table 3.* Same-backbone ablation on COCO-Position. All variants use the same MAR backbone and evaluation protocol.

| Method | AP ↑ | mIoU ↑ |
|---|---|---|
| Plain MAR | 15.20 | 35.10 |
| Layout Concat | 22.30 | 45.60 |
| Cross-Attn | 27.60 | 51.90 |
| Soft Layout Bias | 45.80 | 70.90 |
| Binary $\mathbf{X} \rightarrow \mathbf{L}$ Bias | 51.00 | 74.80 |
| StructMAR w/o GRPO | 52.40 | 75.90 |
| StructMAR (Full) | **57.20** | **79.40** |

*Table 4.* Detector sensitivity of GRPO under fixed GroundingDINO evaluation. For COCO-Position, each entry reports AP/mIoU; for COCO-MIG, each entry reports ISR/mIoU.

| Benchmark | No GRPO | YOLOv8 | Faster R-CNN | GroundingDINO |
|---|---|---|---|---|
| COCO-Position | 52.40/75.90 | 56.14/78.18 | 57.20/79.40 | **57.68/79.76** |
| COCO-MIG | 58.58/54.96 | 60.63/56.21 | 61.70/56.90 | **62.24/57.73** |

*Table 5.* Inference efficiency decomposition. StructMAR uses more refinement steps than MIGC but has substantially lower per-step latency.

| Method | Steps | Per-step latency (ms) ↓ | Time/image (s) ↓ | Speedup ↑ |
|---|---|---|---|---|
| MIGC | 50 | 312.2 | 15.61 | 1.00× |
| Plain MAR | 64 | 55.6 | 3.56 | 4.39× |
| StructMAR w/o GRPO | 64 | 59.1 | 3.78 | 4.13× |
| StructMAR (Full) | 64 | 60.2 | 3.85 | 4.05× |

rather, it mainly comes from the lower per-step latency of masked autoregressive decoding. We provide a detailed efficiency decomposition in Sec. 4.4.

### 4.4. Ablation and Robustness Analysis

Table 3 isolates the contribution of each component under the same MAR backbone. The proposed binary $X \rightarrow L$ layout-attention bias provides the largest gain, and GRPO further improves the final model through metric-aligned refinement.

Table 4 evaluates GRPO with different reward detectors while keeping GroundingDINO as the fixed evaluator. All reward-detector variants improve over the no-GRPO model, suggesting that the gains are not tied to a single reward detector. Faster R-CNN provides most of the improvement and is used for our main results.

**Reward decomposition and detector bias.** We further decompose the GRPO reward into IoU, confidence, and centroid terms. The main improvement is driven by the IoU term, while the centroid term provides additional geometric stabilization, especially in dense layouts. The confidence term alone is less reliable because it can favor detector-visible objects without necessarily improving box-level ge-

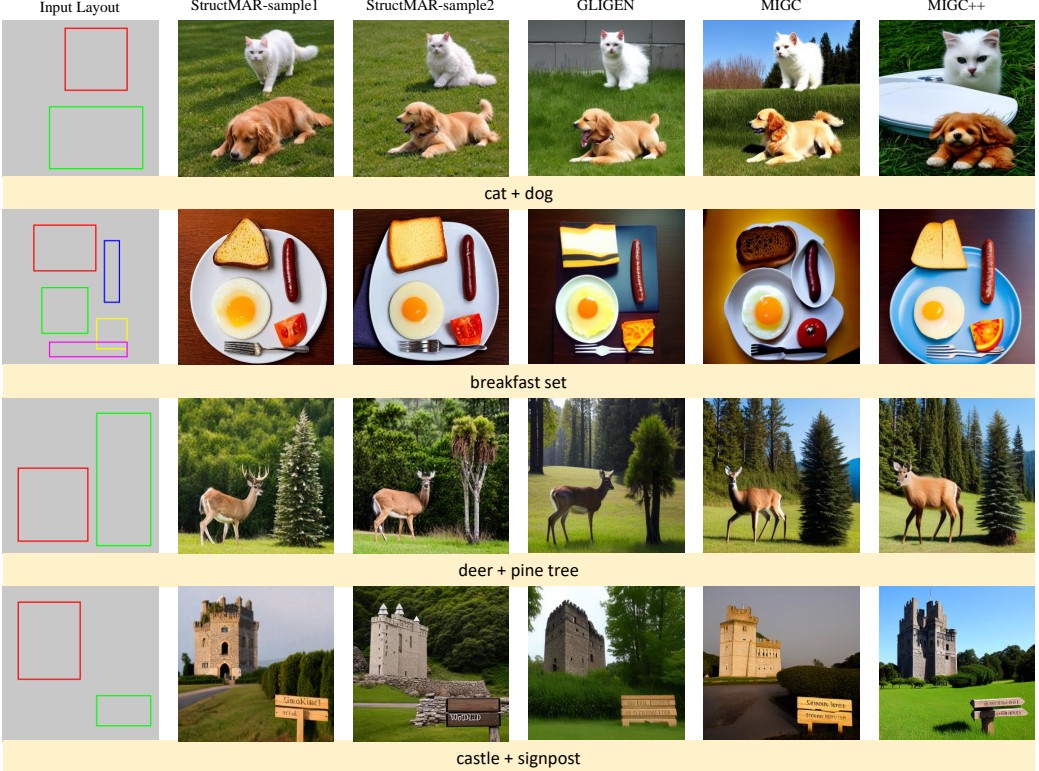

*Figure 3.* **Qualitative comparison under the same input layouts.** Each row shows one text prompt and its corresponding input layout. The first column visualizes the input bounding-box layout, the next two columns show two samples generated by StructMAR, and the remaining columns show results from GLIGEN, MIGC, and MIGC++. MIGC++ is included only as a qualitative reference. StructMAR more consistently places instances inside their target regions, especially in crowded or overlap-heavy scenes where soft layout conditioning can be ambiguous.

ometry. Thus, the detector-induced bias mainly encourages tighter and more spatially consistent object localization. This bias is aligned with the layout-execution objective, but it may also favor geometrically regular placements; we discuss this limitation in Sec. 6.

The speedup is not due to fewer refinement iterations: StructMAR uses 64 masked refinement steps, whereas MIGC uses 50 denoising steps. The main source is lower per-step latency of masked autoregressive decoding. The structure-aware modules introduce only modest overhead relative to Plain MAR, preserving most of the efficiency advantage of the backbone.

## 5. Conclusion

We present **StructMAR**, a structure-aware masked autoregressive framework for layout-controlled text-to-image generation. By combining 2D RoPE, layout-guided attention-logit bias, gated text cross-attention, and GRPO-based metric refinement, StructMAR strengthens latent-to-layout correspondence while preserving semantic coherence and efficient decoding. Experiments on COCO-Position and COCO-

MIG show improved layout fidelity, robustness under dense layouts, and a $4.05\times$ inference speedup over MIGC. These results suggest that explicit attention-level structural bias is a promising direction for efficient and spatially faithful controllable generation.

## 6. Limitations

StructMAR focuses on bounding-box-conditioned generation and uses detector-based GRPO rewards, which may inherit detector-specific biases toward clear object boundaries or regular layouts. Detector-based optimization also cannot fully capture perceptual quality or human preference. Extending the framework to masks, free-form layouts, noisier annotations, broader domains, fine-grained attributes, and human-preference evaluation remains future work.

## Impact Statement

This work improves layout-conditioned text-to-image generation, with potential benefits for creative design, visual content production, and synthetic data generation. It may also increase the risk of misleading synthetic content, so re-

sponsible deployment, provenance tracking, and safeguards against misuse and detector-induced biases are encouraged.

## Acknowledgements

This work was supported by High-Performance Computing Platform of Xidian University.

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

# A. Additional Method Details

## A.1. Latent Grid, Coordinate System, and Box-to-Token Mapping

**Latent tokenization.**   We encode an image $\mathbf{I} \in \mathbb{R}^{H_0 \times W_0 \times 3}$ into a continuous latent grid using a frozen VAE encoder $E$:

$$\mathbf{Z} = E(\mathbf{I}) \in \mathbb{R}^{H \times W \times d}, \quad H = \frac{H_0}{s}, \; W = \frac{W_0}{s}, \tag{14}$$

where $s$ is the VAE downsampling factor (e.g., $s = 8$ for common latent VAEs). We flatten $\mathbf{Z}$ into a token sequence $\mathbf{X} = \{\mathbf{x}_i\}_{i=1}^{HW}$ with a deterministic index mapping:

$$i = u + v \cdot W, \quad u \in \{0, \ldots, W - 1\}, \; v \in \{0, \ldots, H - 1\}. \tag{15}$$

**Coordinate normalization.**   For RoPE and box membership tests, each token has a 2D coordinate $(u_i, v_i)$.

**Box normalization.**   Each box $b_j = (x_1, y_1, x_2, y_2) \in [0, 1]^4$ is normalized in the image coordinate system. We map boxes to latent grid coordinates via:

$$\begin{aligned} u_1^{(j)} &= \lfloor x_1 W \rfloor, \quad u_2^{(j)} = \lceil x_2 W \rceil - 1, \\ v_1^{(j)} &= \lfloor y_1 H \rfloor, \quad v_2^{(j)} = \lceil y_2 H \rceil - 1. \end{aligned} \tag{16}$$

A token $(u_i, v_i)$ is considered inside $b_j$ if and only if:

$$u_1^{(j)} \leq u_i \leq u_2^{(j)} \quad \text{and} \quad v_1^{(j)} \leq v_i \leq v_2^{(j)}. \tag{17}$$

**Edge cases.**   For tiny boxes that collapse under downsampling (e.g., mapping to an empty region), we apply a *minimal coverage rule*: we expand the mapped box to include at least one token (the nearest token to the box center), ensuring every instance has a valid supervision/constraint target.

## A.2. 2D RoPE Formulation for Flattened Latent Tokens

Our transformer operates on a flattened sequence but uses 2D rotary positional embeddings to preserve relative 2D topology. Let the head dimension be $d_h$, and define $d_h = 2d_r$ (since RoPE requires pairing). We allocate two independent rotary components for the horizontal and vertical coordinates:

- The first half of channels encode $u$ ($x$-axis).

- The second half encode $v$ ($y$-axis).

For token $i$ with query/key projections $\mathbf{q}_i, \mathbf{k}_i \in \mathbb{R}^{d_h}$, we split them as:

$$\mathbf{q}_i = [\mathbf{q}_i^u; \mathbf{q}_i^v], \quad \mathbf{k}_i = [\mathbf{k}_i^u; \mathbf{k}_i^v], \quad \text{where } \mathbf{q}_i^u, \mathbf{k}_i^u \in \mathbb{R}^{d_r}. \tag{18}$$

We then apply 1D RoPE separately:

$$\tilde{\mathbf{q}}_i^u = \text{RoPE}(\mathbf{q}_i^u, u_i), \quad \tilde{\mathbf{q}}_i^v = \text{RoPE}(\mathbf{q}_i^v, v_i), \tag{19}$$

and similarly for $\tilde{\mathbf{k}}_i$. The final embedding is obtained by concatenation:

$$\tilde{\mathbf{q}}_i = [\tilde{\mathbf{q}}_i^u; \tilde{\mathbf{q}}_i^v], \quad \tilde{\mathbf{k}}_i = [\tilde{\mathbf{k}}_i^u; \tilde{\mathbf{k}}_i^v]. \tag{20}$$

## A.3. Layout Token Encoding $f_{\text{layout}}$

Each instance condition $(b_j, c_j)$ is encoded into a layout token $\boldsymbol{\ell}_j$.

**Geometry embedding.** We embed box geometry with an MLP:

$$\mathbf{g}_j = \mathrm{MLP}([x_1, y_1, x_2, y_2, x_c, y_c, w, h]), \tag{21}$$

where $x_c = (x_1 + x_2)/2$, $y_c = (y_1 + y_2)/2$, $w = x_2 - x_1$, and $h = y_2 - y_1$.

**Category/text embedding.**

- If $c_j$ is a category ID, we use a learnable embedding table $e(c_j)$.

- If $c_j$ is a text tag, we tokenize it and pool (e.g., mean-pooled text encoder output) into $\mathbf{t}_j$.

**Fusion.** The final layout token is computed as:

$$\boldsymbol{\ell}_j = \mathrm{LN}(\mathbf{W}[\mathbf{g}_j; \mathbf{t}_j]), \tag{22}$$

optionally combined with a learned "instance type" embedding indicating a bbox-conditioned token.

**Permutation invariance.** We randomize the order of $\{\boldsymbol{\ell}_j\}$ during training to avoid order leakage, which is recommended for grounding and layout tasks.

## A.4. Layout-Guided Attention Bias as Block-wise Logit Injection

### A.4.1. BLOCK STRUCTURE

We append a background layout token $\boldsymbol{\ell}_0$ and form the extended layout-token sequence $\bar{\mathbf{L}} = [\boldsymbol{\ell}_0; \mathbf{L}]$, where $\bar{\mathbf{L}} \in \mathbb{R}^{(N+1) \times d}$. The joint sequence is $[\bar{\mathbf{L}}; \mathbf{X}]$, with $\mathbf{X} \in \mathbb{R}^{HW \times d}$. The attention logits after 2D RoPE on latent queries/keys are partitioned as:

$$\mathbf{A} = \frac{\tilde{\mathbf{Q}}\tilde{\mathbf{K}}^\top}{\sqrt{d_h}} = \begin{bmatrix} \mathbf{A}_{LL} & \mathbf{A}_{LX} \\ \mathbf{A}_{XL} & \mathbf{A}_{XX} \end{bmatrix}, \tag{23}$$

where the layout block $\mathbf{L}$ denotes the extended sequence $\bar{\mathbf{L}}$ for simplicity. We inject a layout bias $\mathbf{M}^{\mathrm{layout}} \in \mathbb{R}^{HW \times (N+1)}$ only into the bipartite block $\mathbf{A}_{XL}$:

$$\mathbf{A}' = \begin{bmatrix} \mathbf{A}_{LL} & \mathbf{A}_{LX} \\ \mathbf{A}_{XL} + \mathbf{M}^{\mathrm{layout}} & \mathbf{A}_{XX} \end{bmatrix}, \quad \mathrm{Attn} = \mathrm{Softmax}(\mathbf{A}')\mathbf{V}. \tag{24}$$

This restricts layout-token candidates within the $\mathbf{X} \to \mathbf{L}$ block while keeping $\mathbf{A}_{XX}$ intact for global latent-to-latent coherence.

### A.4.2. MASK DEFINITION AND NUMERICAL STABILITY

For a latent token at spatial coordinate $(u_i, v_i)$, the layout bias is defined over the extended layout-token sequence:

$$\mathbf{M}_{i,j}^{\mathrm{layout}} = \begin{cases} 0, & j > 0 \text{ and } (u_i, v_i) \in \mathrm{Interior}(b_j), \\ 0, & j = 0 \text{ and } (u_i, v_i) \notin \bigcup_{k=1}^{N} \mathrm{Interior}(b_k), \\ -\infty, & \text{otherwise.} \end{cases} \tag{25}$$

In practice, $-\infty$ is implemented as a large negative constant $-C$ before softmax to avoid numerical issues.

### A.4.3. TOKENS OUTSIDE ANY BOX

For latent tokens outside all bounding boxes, only the background layout token $\boldsymbol{\ell}_0$ is allowed in the $\mathbf{X} \to \mathbf{L}$ block. For tokens covered by at least one box, $\boldsymbol{\ell}_0$ is masked out by default. This avoids degenerate behavior for background regions while preserving explicit instance-level compatibility.

### A.4.4. OVERLAPPING BOXES AND MULTI-MEMBERSHIP

If two boxes overlap, a token may belong to multiple instances. We use a multi-allow strategy, assigning bias $0$ to all compatible layout tokens. This preserves multiple valid layout candidates in overlapping regions, while GRPO further helps resolve remaining geometric conflicts during metric-aligned refinement.

## A.5. Semantic Safeguard: Gated Cross-Attention to Text

To prevent text semantics from dominating the structure-aware layout pathway, we inject text via a gated residual cross-attention:

$$\mathbf{h} \leftarrow \mathbf{h} + g \cdot \text{CrossAttn}(\mathbf{h}, \mathbf{S}), \quad g = \tanh(\gamma). \tag{26}$$

We initialize $\gamma = 0 \Rightarrow g \approx 0$, so the model initially relies more on the structural layout pathway. As training proceeds, $g$ can adapt to incorporate semantics when it does not conflict with layout guidance.

## A.6. GRPO Fine-tuning Details (Stage II)

We treat the full masked decoding trajectory as a stochastic policy $\pi_\theta(\tau|\mathbf{S}, \mathcal{B})$.

### A.6.1. REWARD DEFINITION

For each generated image $\hat{\mathbf{I}}$, we run a frozen detector (e.g., GroundingDINO or YOLOv8) and compute:

- $\text{IoU}_{\text{det}}$: matching IoU.

- $\text{Conf}_{\text{det}}$: mean matched confidence.

- $\text{Dist}_{\text{centroid}}$: mean centroid distance.

The total reward is defined as:

$$R = \lambda_{\text{IoU}} \text{IoU}_{\text{det}} + \lambda_{\text{conf}} \text{Conf}_{\text{det}} - \lambda_{\text{cent}} \text{Dist}_{\text{centroid}}. \tag{27}$$

### A.6.2. GRPO OBJECTIVE

The Group Relative Policy Optimization (GRPO) update is approximated as:

$$\nabla_\theta \mathcal{J}(\theta) \approx \sum_{g=1}^{G} A_g \nabla_\theta \log \pi_\theta(\tau_g|\mathbf{S}, \mathcal{B}). \tag{28}$$

**Stability constraints:** We optionally add a KL penalty to the supervised model $\theta_0$ to prevent reward hacking.

### A.6.3. LOG-PROBABILITY $\log \pi_\theta(\tau)$ IN MAR

In continuous-latent masked decoding, we compute $\log \pi_\theta(\tau)$ as the sum of log-likelihoods of sampled token updates:

$$\log \pi_\theta(\tau) = \sum_t \sum_{i \in m_t} \log \mathcal{N}(\mathbf{x}_i; \mu_{\theta,i}, \sigma^2 \mathbf{I}). \tag{29}$$

# B. Additional Algorithm

Algorithm 1 provides a detailed pseudocode description of the StructMAR training and inference stages, highlighting the specific injection points for the Layout-Guided Attention Bias and the semantic safeguard mechanism.

---

**Algorithm 1** StructMAR Training and Inference

---

1: **Input:** text tokens $\mathbf{S}$; layouts $\mathcal{B}$; VAE $E, D$; backbone $f_\theta$.
2: **Precompute:** instance layout tokens $\mathbf{L} \leftarrow f_{\text{layout}}(\mathcal{B})$; append background token $\ell_0$ to form $\bar{\mathbf{L}} = [\ell_0; \mathbf{L}]$; build mask $\mathbf{M}^{\text{layout}} \leftarrow \text{BUILDMASK}(\mathcal{B}, H, W)$.
3: *// Stage I: Supervised Learning*
4: **for** each supervised step **do**
5:     $\mathbf{Z} \leftarrow E(\mathbf{I})$; flatten to $\mathbf{X}$ with coords $(u_i, v_i)$
6:     sample mask set $m$ and noise level $t$; construct noised masked tokens $\mathbf{X}_m^{(t)}$
7:     **StructMAR forward:**
8:         apply 2D RoPE to latent-token $\mathbf{Q}_X, \mathbf{K}_X$ using $(u_i, v_i)$
9:         compute attention logits on $[\bar{\mathbf{L}}; \mathbf{X}]$ and inject $\mathbf{M}^{\text{layout}}$ only to block $\mathbf{A}_{X\bar{L}}$
10:         apply gated text cross-attention as semantic safeguard
11:     predict noise $\hat{\epsilon} \leftarrow f_\theta(\mathbf{X}_m^{(t)}, \mathbf{X}_{\neg m}, \mathbf{S}, \bar{\mathbf{L}}; \mathbf{M}^{\text{layout}})$
12:     update $\theta$ by minimizing $\mathcal{L}_{\text{MAR}} = \sum_{i \in m} \|\epsilon_i - \hat{\epsilon}_i\|^2$
13: **end for**
14: *// Stage II: GRPO Fine-tuning*
15: **if** GRPO enabled **then**
16:     **for** each GRPO iteration **do**
17:         sample $G$ trajectories by masked decoding with $f_\theta(\cdot; \bar{\mathbf{L}}, \mathbf{M}^{\text{layout}})$
18:         compute detector-based rewards $\{R_g\}$ and group advantages $\{A_g\}$
19:         update $\theta$ with GRPO objective, optionally with KL regularization to $\theta_0$
20:     **end for**
21: **end if**
22: **Inference:** iterative masked decoding with $f_\theta(\cdot; \bar{\mathbf{L}}, \mathbf{M}^{\text{layout}})$; output $\hat{\mathbf{I}} = D(\hat{\mathbf{Z}})$

---

# C. Experimental Reproducibility Details

## C.1. Training Configuration

For completeness, Table 6 summarizes the key implementation settings used for the main StructMAR experiments.

*Table 6.* Key training and inference configuration for StructMAR.

| Item | Setting |
|------|---------|
| Dataset | COCO 2014 train split with captions, instance labels, and bounding boxes |
| Image size | $256 \times 256$ |
| VAE | Frozen KL-VAE checkpoint `kl16.ckpt` |
| VAE latent setting | Embed dim $= 16$, stride $= 16$, patch size $= 1$ |
| Text encoder | Local CLIP text encoder, `clip-vit-base-patch32` |
| Text representation | Token-level CLIP embeddings for gated cross-attention |
| Backbone | `mar_huge` initialized from ImageNet-pretrained MAR weights |
| Diffusion-loss head | `diffloss_d`=12, `diffloss_w`=1536 |
| Maximum layout instances | 16 objects per image |
| Structural modules | 2D RoPE, layout-guided attention bias, gated text cross-attention |
| 2D RoPE | Enabled, with RoPE base 10000 |
| Text cross-attention | Enabled at every transformer layer |
| Layout-guided bias | Enabled in both encoder and decoder attention blocks |
| Layout bias value | 10000.0 before conversion to a negative masking bias |
| Learned position embedding | Disabled when 2D RoPE is used, unless explicitly retained |
| Label drop probability | 0.1 during supervised layout-conditioned training |
| Stage-I trainable modules | Text-conditioning and layout-related modules |
| Stage-II trainable modules | Full StructMAR model |
| Supervised batch size | 64 |
| Supervised optimizer | AdamW with $\beta_1 = 0.9$, $\beta_2 = 0.95$ |
| Supervised base learning rates | $1 \times 10^{-4}$ for adapter/layout alignment and $2 \times 10^{-5}$ for full SFT |
| Weight decay in supervised training | 0.0 |
| EMA rate | 0.9999 |
| Gradient clipping in supervised training | 3.0 |
| Gradient checkpointing | Enabled during full SFT |
| RL fine-tuning method | Group Relative Policy Optimization (GRPO) |
| RL initialization | Stage-II supervised checkpoint |
| Reward detector | Faster R-CNN ResNet50-FPN v2 trained on COCO |
| Reward terms | Detection IoU, detection confidence, and centroid-distance penalty |
| GRPO group size | 4 samples per prompt/layout condition |
| RL batch size | 4 |
| GRPO training iterations | 256 update iterations |
| Inference refinement steps | 64 masked refinement steps |
| RL mixture ratio | `rl_step_ratio`=1.0 |
| KL coefficient | 0.02 against the supervised reference model |
| RL learning rate | $1 \times 10^{-6}$ |
| RL weight decay | 0.05 |
| RL advantage normalization | Group-wise z-score normalization |
| Reward clipping | $[-10, 10]$ |
| Gradient clipping in RL | 1.0 |
| Evaluation detector | GroundingDINO, following the MIGC evaluation protocol |
| Sampling CFG | Linear CFG schedule; rollout CFG set to 1.0 during RL |
| Sampling temperature | 1.0 |
| Runtime device | CUDA GPU |
| Implementation precision | Mixed precision is supported through AMP; TF32 can be enabled on compatible GPUs |

**COCO-MIG attribute-aware evaluation.** For COCO-MIG, we use the colored instance descriptions provided by the benchmark as instance text tags. During evaluation, we follow the MIGC protocol: after category and position matching, the color attribute is checked for each matched instance, and an instance with an incorrect color attribute receives IoU 0. Therefore, the reported ISR and mIoU in Table 2 evaluate joint category, position, color, and instance-count correctness

rather than layout-only alignment.

## D. Additional Broader Validation Results

We provide an additional evaluation beyond the main COCO-Position and COCO-MIG protocols. This experiment tests whether the proposed structural execution mechanism can faithfully execute compositional layout plans under a shared prompt-to-layout planner.

Table 7 reports GenEval v1 results under a shared prompt-to-layout planner. This setting evaluates compositional plan execution rather than end-to-end layout planning.

*Table 7.* GenEval v1 under a shared prompt-to-layout planner. This setting evaluates compositional plan execution rather than end-to-end layout planning.

| Method | Overall ↑ | Counting ↑ | Position ↑ |
|---|---|---|---|
| MIGC + planner | 0.69 | 0.72 | 0.75 |
| StructMAR w/o GRPO + planner | 0.72 | 0.77 | 0.81 |
| StructMAR (Full) | **0.75** | **0.80** | **0.85** |

## E. Additional Robustness Results

We further evaluate robustness to noisy or coarse bounding boxes without retraining. As shown in Table 8, StructMAR degrades smoothly under moderate center jitter, box scaling, and aspect distortion, while maintaining a positive margin over MIGC in both AP and mIoU.

*Table 8.* Robustness to noisy or coarse bounding boxes on COCO-Position.

| Perturbation | Severity | MIGC AP | StructMAR AP | MIGC mIoU | StructMAR mIoU |
|---|---|---|---|---|---|
| None | 0 | 54.69 | 57.20 | 77.38 | 79.40 |
| Center jitter | 5% | 53.36 | 55.96 | 76.18 | 78.12 |
| Center jitter | 10% | 50.98 | 54.18 | 73.82 | 76.28 |
| Box expand/shrink | 10% | 51.44 | 54.36 | 74.16 | 76.46 |
| Aspect distortion | 10% | 50.62 | 53.04 | 73.34 | 75.58 |

# F. Additional Qualitative Results and Failure Modes

This appendix provides supplementary qualitative results to complement the main-paper figures. We summarize representative successful cases in this section and discuss failure cases and limitations in Appendix G. Each panel contains 16 subfigures, with the input layouts overlaid as bounding boxes and tags.

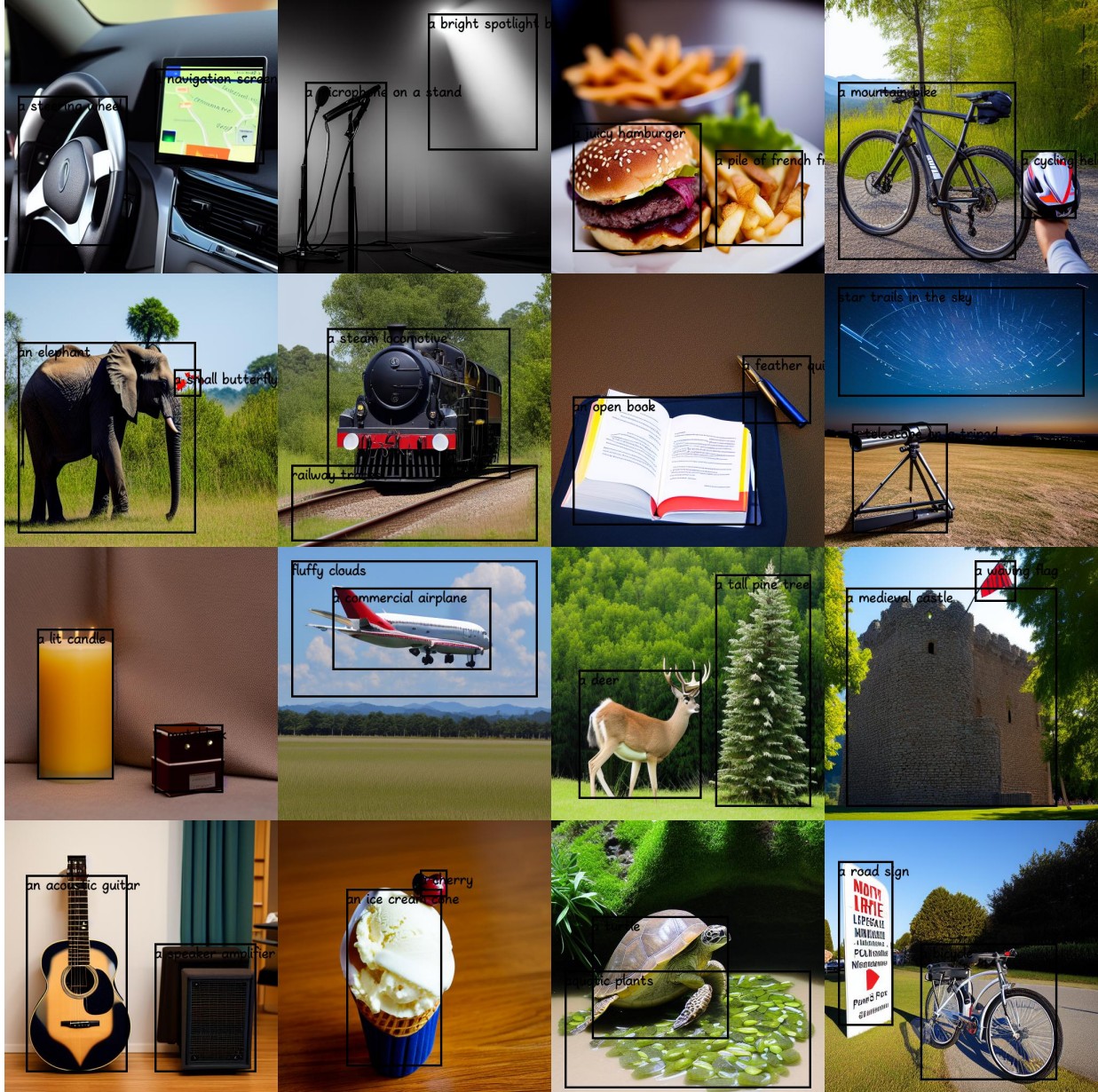

*Figure 4.* **Successful cases: diverse multi-instance scenes under varied scale and context.** This panel includes 16 examples with broader scene diversity and scale variation, such as in-car navigation with the dashboard, a studio setup (microphone stand and spotlight), food pairs (burger/fries), bicycle with accessories, wildlife with small companion objects (elephant/butterfly), locomotives, book-and-pen tabletop scenes, night-sky star trails with a telescope, airplanes with clouds, deer with trees, castles with flags, guitars with amplifiers, ice-cream close-ups, turtles with aquatic plants, and road signs with bicycles. In many samples, multiple instances appear in their designated regions with limited interference, indicating stable layout adherence across heterogeneous backgrounds.

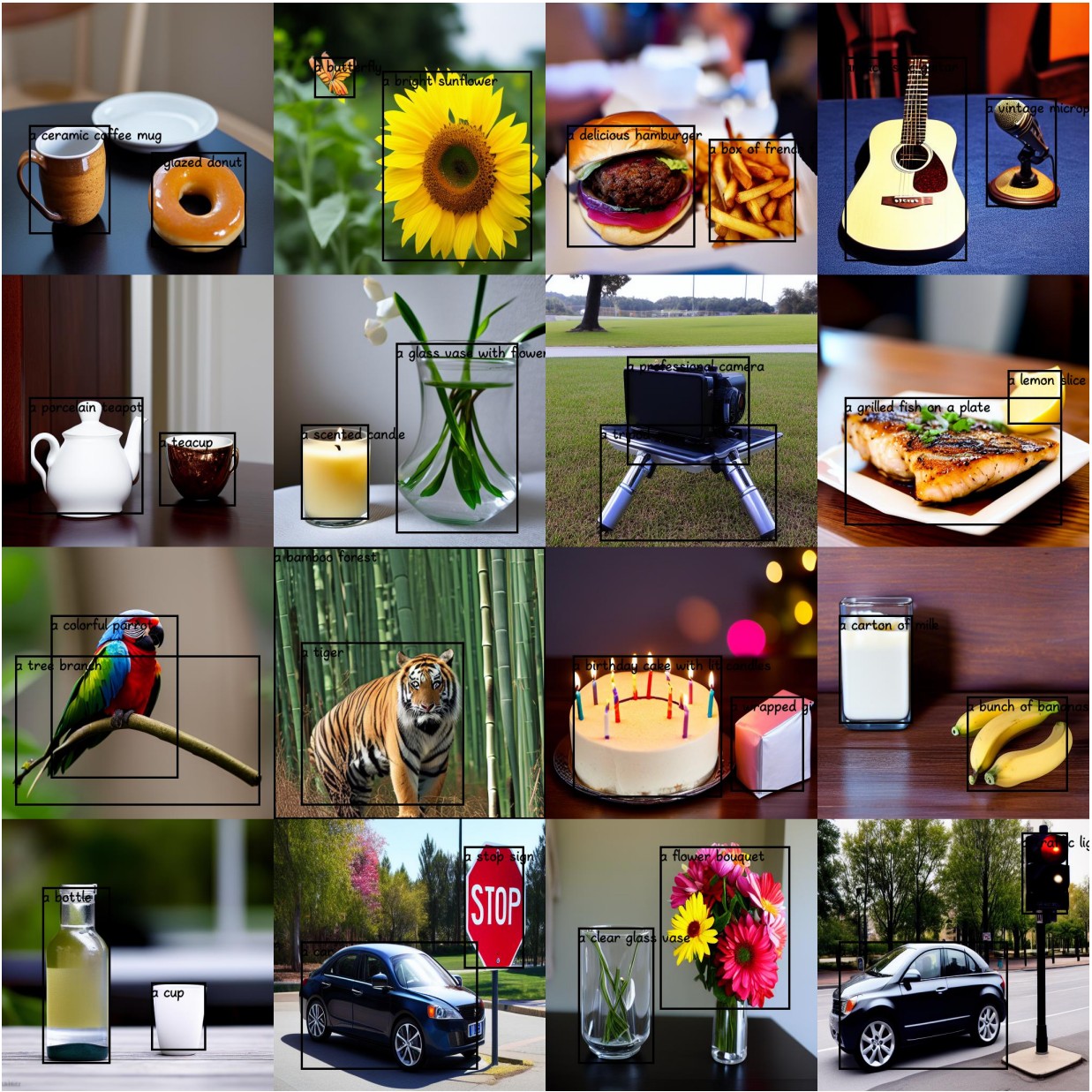

*Figure 5.* **Successful cases: attribute- and category-diverse object compositions.** This panel presents 16 object-centric compositions emphasizing category diversity and co-occurrence, including mug+donut, sunflower(+small insect), burger+fries, guitar+microphone, teapot+teacup, candle+glass vase, camera-on-tripod scenes, grilled fish+lemon slice, parrot-on-branch, tiger-in-bamboo, birthday cake+wrapped gift, milk carton+bananas, bottle+cup, car+stop sign, bouquet+glass vase, and car+traffic light. The instances generally appear within the annotated boxes, with coherent object–background interactions in many cases (e.g., tabletop lighting, outdoor depth cues).

# G. Failure Cases and Qualitative Guidance

We further include representative failure cases to clarify limitations under strict layout control and multi-instance constraints. Each panel contains 16 subfigures.

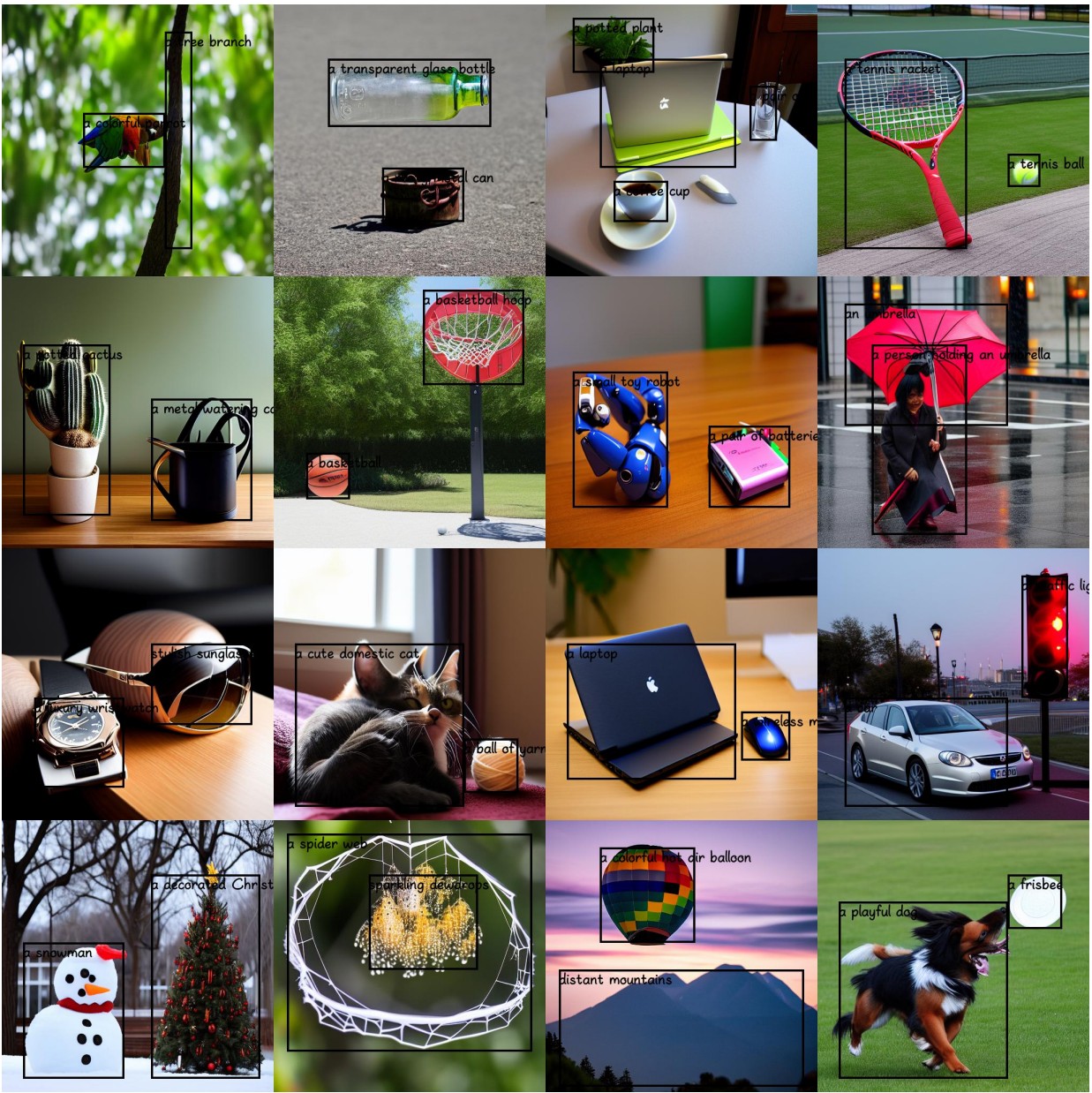

*Figure 6.* **Failure Case Figure 1.** This collage shows representative failures that mainly occur for small or low-salience targets. Although objects are often generated near the specified regions, some small instances (e.g., small balls or plants) appear weakly expressed or incomplete. For visually subtle categories (e.g., transparent containers) or thin-structure patterns (e.g., web-like regions), the model may produce ambiguous boundaries within the box. We also observe occasional minor position drift and limited boundary spillover in multi-instance scenes.

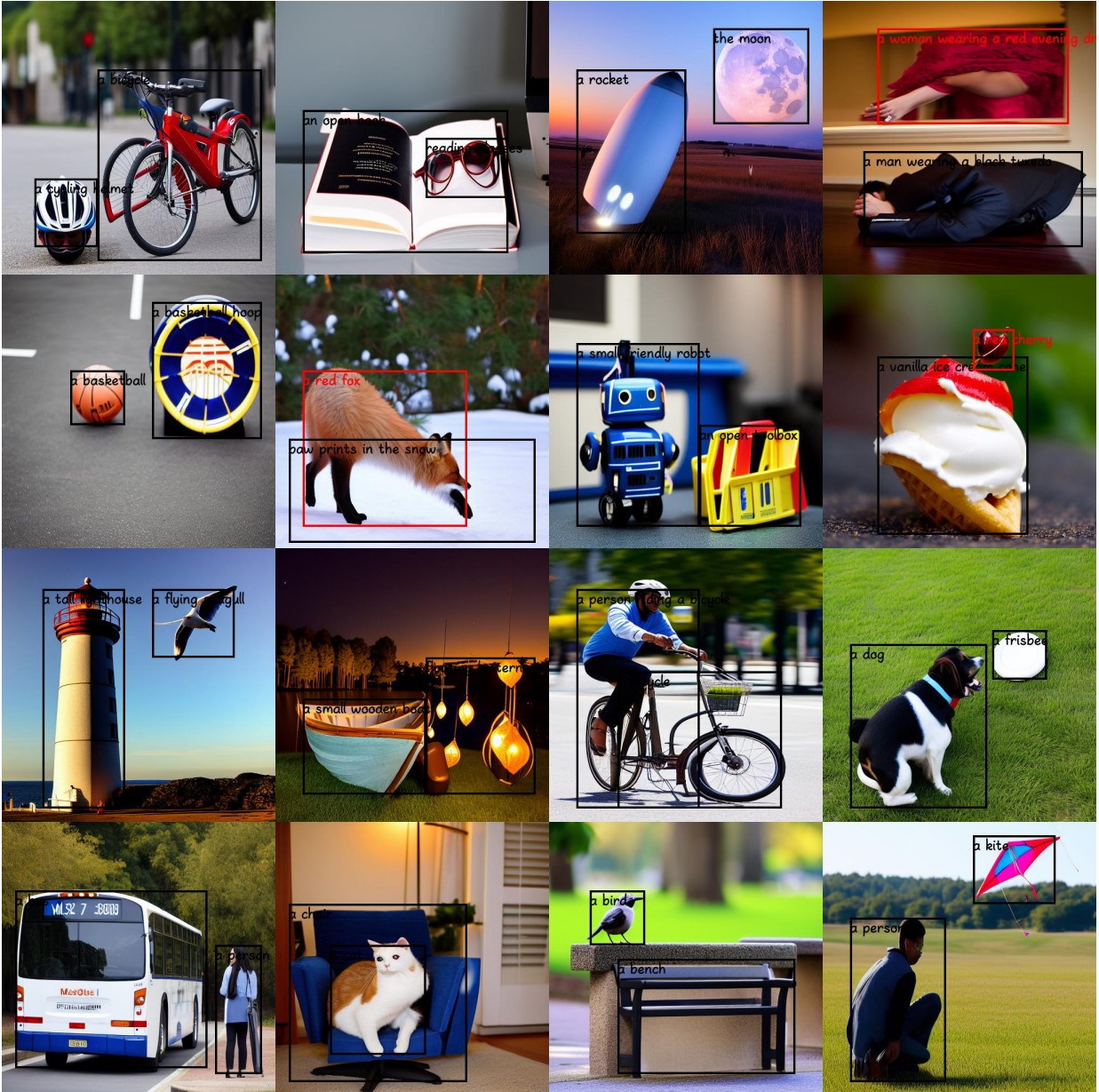

*Figure 7.* **Failure Case Figure 2.** This collage summarizes additional layout-adherence failures under more challenging semantics and composition. For attribute-dependent or human-centric conditions, the model can yield partial or proxy renderings inside the target box, resulting in semantic mismatch. For auxiliary or fine-grained targets (e.g., small markings), the intended instance may be missing or absorbed into context. With multiple boxes, instance competition may lead to scale mismatch, partial occlusion/compression, or generation of a related but not fully matching object relative to the specified box and tag.

