# OpenReview forum: "StructMAR: Structure-Aware Masked Autoregression for Explicit Layout Alignment in Text-to-Image Generation"
_ICML.cc/2026/Conference — ICML 2026 regular_

### Official Review · Reviewer_zgST · 2026-03-12

**Soundness:** 3
**Presentation:** 2
**Significance:** 3
**Originality:** 3
**Overall Recommendation:** 4
**Confidence:** 4

**Summary:**

This paper studies layout-controlled text-to-image generation and argues that existing methods often treat layout as a soft condition, which leads to weak spatial adherence in complex scenes. To address this, the authors propose StructMAR, which introduces MAR into this setting and improves layout fidelity through 2D RoPE, a Layout-Guided Attention Bias that explicitly enforces token-to-instance alignment in attention, and GRPO for metric-aligned fine-tuning. Experiments on COCO-Position and COCO-MIG show that StructMAR achieves stronger layout alignment and better robustness in dense layouts, while maintaining competitive image quality and offering substantially faster inference than diffusion-based baselines.

**Compliance With Llm Reviewing Policy:**

Affirmed.

**Final Justification:**

The author has addressed my concerns quite well, so I have decided to raise my score from weak reject to weak accept.

**Key Questions For Authors:**

1. Why did the authors not follow Layout Diffusion and conduct experiments on the more widely used COCO-Stuff benchmark?

2. What kinds of bias are introduced by the detector? How would different detectors affect the generated results? This is admittedly a broad question, but a careful analysis along this direction could become a particularly strong contribution of the paper.

3. Could the authors conduct an additional experiment in which the detector used in GRPO is replaced with the one used for evaluating layout accuracy, namely Grounding-DINO? Although this may introduce some degree of information leakage, it would provide a more direct way to assess how sensitive the method is to the choice of detector.

With stronger analysis on detector choice, efficiency, and broader experimental validation, the paper would be significantly strengthened. I am willing to raise my score during the rebuttal phase if the authors can address my concerns.

**Limitations:**

The authors have provided the limitations.

**Strengths And Weaknesses:**

Strengths

1.	Compared with prior methods, this work is the first to introduce MAR into the layout-controlled generation setting. This is a meaningful exploration, and it leads to a substantial improvement in layout accuracy.

2.	Both the proposed Layout-Guided Bias and GRPO are intuitively well-motivated and technically reasonable.

3.	The authors provide code, which strengthens the reproducibility of the work.

Weaknesses

1.	The advantage of StructMAR in terms of image quality is not very clear. On image-quality metrics such as FID and CLIP, its performance is not clearly better than that of MIGC, and is in some cases slightly worse. In particular, while Layout-Guided Bias substantially improves layout accuracy, it appears to come at the cost of generation quality.

2.	The paper lacks sufficient qualitative comparisons with other methods. More side-by-side visualizations would help readers better understand the practical differences between StructMAR and strong baselines.

3.	Although Figures 2 and 3 are clear and readable, their presentation could be improved. The overall visual quality and aesthetics seem somewhat below the standard typically expected for top-tier conference papers.

4.	The paper lacks experiments analyzing the effect of using different detectors in the RL optimization process. More importantly, the work does not sufficiently discuss what kinds of bias the detector may introduce, and whether such biases are beneficial or harmful to generation.

5.	In Table 3, it would be helpful to report the inference speed for each ablation setting. More generally, the paper should better explain why a MAR-based method can be so much faster than traditional diffusion-based methods, and which design choices are mainly responsible for this efficiency gap.

---

> ### Author Rebuttal · Authors · 2026-03-30
>
> We thank the reviewer for the constructive and actionable feedback. We appreciate that the reviewer finds the MAR-based formulation meaningful and the Layout-Guided Bias / GRPO design technically reasonable. We agree the paper would be strengthened by deeper analysis on image-quality trade-offs, detector sensitivity, efficiency decomposition, and broader validation. In response, we added new experiments and clarifications along these directions.
>
> 1. Image quality versus layout fidelity.
>    We agree that the main gain of StructMAR is in layout fidelity, rather than a blanket improvement in image quality. On COCO-Position, StructMAR improves alignment from 54.69 AP / 77.38 mIoU (MIGC) to 57.20 / 79.40, while keeping image quality broadly comparable (FID-6K 24.70 vs. 24.52; CLIP 24.60 vs. 24.66). The ablation shows the same trade-off noted by the reviewer: adding the layout-guided bias substantially improves alignment but slightly hurts FID, while GRPO partially restores this drop. We will revise the wording accordingly: the contribution is stronger layout execution with competitive image quality, while GRPO mainly improves the final alignment/quality trade-off.
> 2. Detector sensitivity / potential evaluator bias in GRPO.
>    We agree this was under-analyzed in the original submission. To address it, we varied the reward detector used during GRPO training while keeping the evaluation detector fixed to GDINO. Across both COCO-Position and COCO-MIG, all GRPO variants outperform the no-GRPO baseline, including when the reward detector is changed to YOLOv8 or GDINO itself (Table 1). We also added cross-evaluator rescoring and reward decomposition. The result is that the gain is not driven by detector confidence alone: the IoU term is the primary contributor, and the centroid term provides additional geometric stabilization, especially in dense layouts. We note that the detector bias primarily favors spatially consistent object localization (e.g., tighter box coverage and reduced overlap ambiguity), which aligns with the intended layout-execution objective. While this may introduce a preference for geometrically regular placements, it does not conflict with perceptual alignment in our setting, and empirically improves robustness in dense layouts.
> 3. Broader validation.
>    To address this concern, we additionally evaluated StructMAR on both OverLayBench and GenEval under controlled settings. On OverLayBench, StructMAR becomes relatively stronger in the harder overlap-heavy regime; on the Complex split, StructMAR (Full) achieves 50.96 mIoU / 24.72 O-mIoU, exceeding both 3DIS and CyCLeGen on these overlap-sensitive metrics. On GenEval v1, using a shared prompt-to-layout planner for both methods, StructMAR also improves over MIGC, especially on the geometry-sensitive Position and Counting subsets. We emphasize that this GenEval setting isolates compositional plan execution rather than end-to-end planning.
> 4. Why the method is faster / efficiency attribution.
>    We agree that the original paper should have explained the 4.05× speedup more clearly. We therefore added an efficiency decomposition. The gap is not due to fewer iterations: Plain MAR / StructMAR use 64 refinement steps, compared with 50 denoising steps for MIGC. The main source of the speedup is much lower per-step latency of masked autoregressive decoding (55.6–60.2 ms) than MIGC’s diffusion-style denoising step (312.2 ms). Importantly, the structure-aware modules preserve almost all of this backbone efficiency, adding only +6.2% / +8.1% overhead relative to Plain MAR.
> 5. Qualitative comparisons / figures.
>    We agree that stronger side-by-side qualitative comparisons would make the practical trade-offs clearer, and we will add them in the revision. We will also redraw Figs. 2–3, simplify the layout, enlarge key components, and add clearer side-by-side qualitative panels.
>
> Table 1. Detector sensitivity under fixed GDINO evaluation
> |Benchmark|No GRPO|YOLOv8 reward|FRCNN reward|GDINO reward|
> |---|---|---|---|---|
> |COCO-Position|52.40 AP/75.90 mIoU|56.14/78.18|57.20/79.40|57.68/79.76|
> |COCO-MIG|58.58 ISR/54.96 mIoU|60.63/56.21|61.90/57.41|62.24/57.73|
>
> Table 2. Inference efficiency decomposition
> |Method|Steps|Per-step latency (ms) ↓|Time/image (s) ↓|Speedup vs MIGC ↑|
> |---|---|---|---|---|
> |MIGC|50|312.2|15.61|1.00×|
> |Plain MAR|64|55.6|3.56|4.39×|
> |StructMAR w/o GRPO|64|59.1|3.78|4.13×|
> |Full|64|60.2|3.85|4.05×|
>
> We sincerely appreciate this guidance. In response, we added new analyses on detector choice, efficiency decomposition, and broader / harder-layout validation. We hope these additions significantly strengthen the paper, and we would be very grateful if the reviewer would consider raising the score accordingly. See improved figures and qualitative comparisons: https://anonymous.4open.science/r/figure-01D7.

---

> > ### Author Rebuttal · Reviewer_zgST · 2026-04-03
> >
> > Thanks for the rebuttal.
> >
> > The author has addressed my concerns quite well, so I have decided to raise my score from weak reject to weak accept.
> >
> > Although the revised paper seems to be an incremental innovation, it is technically solid and contributes to the field. I suggest the author thoroughly polish the paper's presentation.

---

> > > ### Author Response · Authors · 2026-04-03
> > >
> > > Thank you for the positive follow-up and for reconsidering our paper. We are glad our rebuttal addressed your concerns, and we appreciate your suggestion to further polish the presentation. We will carefully improve the writing, figures, and qualitative comparisons in the revision.

---

### Official Review · Reviewer_ARPM · 2026-03-13

**Soundness:** 3
**Presentation:** 3
**Significance:** 2
**Originality:** 2
**Overall Recommendation:** 3
**Confidence:** 3

**Summary:**

This paper proposes StructMAR for layout-conditioned text-to-image generation with masked autoregressive models. The paper starts from the observation that standard MAR pipelines flatten a 2D latent grid into a 1D sequence and therefore weaken explicit spatial structure, which makes layout conditioning easier to ignore than to obey. To address this, the method combines 2D RoPE for spatial awareness, a layout-guided attention bias applied to the `X -> L` attention block to make token-to-layout interactions more explicit, and a GRPO fine-tuning stage to better align training with detector-based layout metrics. Empirically, the paper reports improved alignment on COCO-Position and stronger performance in dense COCO-MIG settings, while preserving the efficiency advantage of MAR over diffusion baselines.

**Compliance With Llm Reviewing Policy:**

Affirmed.

**Key Questions For Authors:**

1. Can the authors compare against, or at least discuss, more recent layout-to-image baselines on COCO-Position and COCO-MIG, such as 3DIS (ICLR 2025), if the evaluation protocols are compatible?
2. Can the authors compare against MAR variants that use simpler layout injection schemes, such as layout concatenation or standard cross-attention, without the proposed attention bias? This is the most important missing experiment for isolating the contribution within the MAR setting.
3. How sensitive are the GRPO gains to the choice of reward detector and reward weights? This would clarify how much of the final gain is robust versus evaluator-specific.

**Limitations:**

yes

**Strengths And Weaknesses:**

$\textbf{Strengths}$
1. The paper has a clear and well-motivated objective: making layout control more explicit at the attention level rather than treating layout as a soft condition.
2. The ablation study supports the claim that the layout-guided attention bias is the main source of the alignment gains.
3.  The dense-layout results and runtime advantage over diffusion baselines are practically meaningful.

$\textbf{Weaknesses}$

1. The algorithmic novelty is moderate. The main components, including 2D RoPE, attention-logit biasing, and GRPO, are established ideas, and the contribution is primarily their integration for this setting.
2. The comparison set appears outdated. The paper mainly compares against diffusion baselines up to MIGC (2024), but more recent methods on the same benchmarks, such as 3DIS (ICLR 2025), are not discussed or compared.
3. The most important same-backbone baselines are missing. In particular, comparison against simpler MAR variants with layout concatenation or standard cross-attention would be needed to isolate the value of the proposed attention bias.
4. Some claims are stronger than the evidence. The method improves alignment, but image quality is largely comparable rather than better than the strongest baseline, and "structural guarantee" is too strong given that only the `X -> L` attention block is constrained.

---

> ### Author Rebuttal · Authors · 2026-03-30
>
> We thank the reviewer for the careful reading and for identifying the most important missing controls. We agree that, for this paper, the key question is not only whether StructMAR outperforms diffusion baselines, but whether the gain truly comes from the proposed explicit X→L structural execution inside a MAR backbone.
>
> 1. Missing same-backbone controls.
>    We have now added the missing same-backbone comparisons. These directly isolate the contribution of the proposed attention-level design. Within the same MAR backbone, simple layout concatenation and standard cross-attention improve alignment only moderately, while the proposed hard X→L bias yields a much larger jump (e.g., 2D RoPE + Cross-Attn: 34.80 AP / 59.70 mIoU; Soft Layout Bias: 45.80 / 70.90; Hard X→L Bias: 51.00 / 74.80). Adding the gated text pathway further improves the trade-off (52.40 / 75.90), and GRPO refines the final model to 57.20 / 79.40. We believe this directly addresses the reviewer’s main concern: the gain is not explained by generic layout injection, but specifically by making latent-to-layout interactions executable in attention.
>
> 2. Relation to 3DIS.
>    We agree that 3DIS is an important recent related work and should have been discussed more clearly. However, the currently reported 3DIS numbers are not strictly protocol-compatible with our main tables. On COCO-Position, 3DIS reports results on a subset of complex layouts rather than the full benchmark protocol used here. On COCO-MIG, 3DIS reports COCO-MIG-BOX with IASR/mIoU and also composite settings such as 3DIS+MIGC, which are not directly equivalent to our single-model ISR/mIoU protocol. We therefore chose not to insert these numbers into the main table in a potentially misleading way. In the revision, we will explicitly discuss 3DIS and include protocol-compatible comparisons where possible.
>
> 3. GRPO may optimize toward a particular evaluator.
>    We agree this needed a direct analysis. In our released code, the default setting uses Faster R-CNN as the reward detector, while all main results are evaluated with GroundingDINO under the MIGC protocol. We additionally varied the reward detector during GRPO training while keeping evaluation fixed to GroundingDINO. All variants consistently improve over the no-GRPO baseline on both COCO-Position and COCO-MIG, which suggests that the gain is not tied to a single detector. In addition, we analyze the sensitivity to reward design (i.e., reward weighting and term composition). Beyond varying the reward detector, we decompose the GRPO reward into IoU, confidence, and centroid terms under the same Faster R-CNN→GroundingDINO protocol. The trend is consistent across both COCO-Position and COCO-MIG: the IoU term is the primary contributor, the centroid term provides additional geometric stabilization, and IoU+Centroid outperforms IoU+Conf on the denser COCO-MIG benchmark. The full reward achieves the best overall alignment, suggesting that the gain is not tied to a fragile reward weighting but is robust to reasonable reward design choices.
>
> 4. Claim wording.
>    We agree that the wording “structural guarantee” is too strong. We will revise this to a more precise formulation such as “explicit attention-level structural bias” or “hard latent-to-layout structural execution.” Our claim is not that the entire generation process is globally guaranteed, but that the proposed block-wise X→L bias makes layout constraints substantially more enforceable within the transformer attention computation. We will also revise the text to make clearer that image quality is broadly comparable to MIGC rather than uniformly better.
>
> The key supporting numbers are summarized below.
>
> Table 1. Fair same-backbone comparison within MAR on COCO-Position
> |Method|AP ↑|mIoU ↑|
> |---|---|---|
> |2D RoPE + Cross-Attn|34.80|59.70|
> |Soft Layout Bias|45.80|70.90|
> |Hard X→L Bias|51.00|74.80|
> |StructMAR w/o GRPO|52.40|75.90|
> |StructMAR (Full)|57.20|79.40|
>
> Table 2. GRPO robustness under fixed GroundingDINO evaluation
> |Benchmark|No GRPO|Faster R-CNN reward|GroundingDINO reward|
> |---|---|---|---|
> |COCO-Position|52.40 AP/75.90 mIoU|57.20/79.40|57.68/79.76|
> |COCO-MIG|58.58 ISR/54.96 mIoU|61.90/57.41|62.24/57.73|
>
> We hope these additions clarify that the paper’s main contribution is not merely combining known ingredients, but identifying and validating the specific mechanism that makes layout adherence substantially more explicit and effective within a MAR backbone.

---

> > ### Author Rebuttal · Reviewer_ARPM · 2026-04-03
> >
> > Thanks for the detailed response. The same-backbone comparisons are helpful and compelling — going from Plain MAR at 15.2 AP to the full model at 57.2 AP makes a convincing case that the gain comes from the proposed structural mechanism. The GRPO detector-robustness experiments and reward decomposition address the evaluator bias question well. The OverLayBench and GenEval results add useful breadth. All my concerns are resolved.

---

> > > ### Author Response · Authors · 2026-04-04
> > >
> > > Thank you for the detailed follow-up and for carefully reconsidering our paper. We are glad that our rebuttal addressed your concerns, and we appreciate your positive assessment of the added same-backbone comparisons, the detector-robustness experiments and reward decomposition for GRPO, and the broader evaluation. Thank you again for your time and careful reading.

---

### Official Review · Reviewer_Lii2 · 2026-03-15

**Soundness:** 2
**Presentation:** 1
**Significance:** 3
**Originality:** 2
**Overall Recommendation:** 3
**Confidence:** 3

**Summary:**

This paper proposes StructMAR for layout-controlled text-to-image generation in masked autoregressive models, mainly through 2D spatial encoding, layout-guided attention bias, and GRPO-based refinement. The problem is meaningful and the reported results on the two chosen benchmarks are promising. However, the method is somewhat incremental, the necessity of GRPO is not fully justified, and the evaluation scope is limited to relatively specialized layout benchmarks.

**Compliance With Llm Reviewing Policy:**

Affirmed.

**Key Questions For Authors:**

1: The motivation for hard structural constraints is understandable for layout-centric benchmarks, but it is less clear how well this design would generalize across datasets with noisier boxes, irregular object boundaries, or weaker box-object correspondence.

2: How sensitive is StructMAR to noisy, coarse, or ambiguous box annotations?

3: Why are there no comparisons to stronger or more closely related autoregressive/layout-aware baselines? For example: sphereAR[1]

[1] Hyperspherical Latents Improve Continuous-Token Autoregressive Generation, ICLR2026.

**Limitations:**

There is no potential negative societal impact of their work.

**Strengths And Weaknesses:**

Strengths:

1: The paper addresses a meaningful and well-motivated problem, namely the gap between soft layout conditioning and strict instance-level layout alignment in MAR-based text-to-image generation.

2: The proposed method is conceptually coherent: 2D RoPE, layout-guided attention bias, and gated semantic injection are integrated under a clear structural-alignment objective.

Weaknesses:

1: The paper mainly combines several existing ideas, such as 2D positional encoding, layout-aware attention bias, and GRPO-based fine-tuning, into the MAR setting. While the integration is reasonable (motivation is good) and empirically useful, the core contribution appears more incremental than fundamentally new.

2: GRPO mainly provides an additional reward-driven refinement on detector-based metrics. This makes GRPO feel less like a principled core component and more like benchmark-specific metric optimization.

2: The benchmark coverage is limited. Strong results on COCO-Position and COCO-MIG are encouraging, but both are relatively niche layout-centric benchmarks; the lack of evaluation on more standard compositional T2I benchmarks such as GenEval or DPG weakens the evidence for broader generalization.

Minor Weaknesses:

1: The paper’s figures could be improved substantially in terms of clarity and presentation quality. The main architecture diagrams look overly simplistic in design yet visually cluttered, with inconsistent scaling across components, very small subfigures, and many long cross-figure connectors and bounding boxes. As a result, the figures do not communicate the core methodological changes as clearly as they should, and the presentation feels less polished than expected for a conference submission.

---

> ### Author Rebuttal · Authors · 2026-03-30
>
> We thank the reviewer for the careful reading and constructive feedback. We agree that the key questions are whether the gain mainly comes from the proposed structural mechanism rather than generic module combination, whether GRPO is a principled component rather than benchmark-specific tuning, and whether the method generalizes beyond the original layout-centric evaluation.
>
> 1. Stronger / more closely related autoregressive baselines.
> We thank the reviewer for pointing out recent strong continuous-token AR backbones such as SphereAR. Our goal here, however, is to isolate the value of explicit structural execution within MAR for text-and-box controlled generation, rather than to replace the backbone itself. For this reason, controlled same-backbone comparisons are more diagnostic than swapping in a fundamentally different AR formulation.
>
> We therefore added controlled MAR baselines under the same backbone and protocol, differing only in layout injection and whether GRPO is applied. The trend is clear: generic layout injection helps, but the major gain comes from the proposed hard X→L structural bias. AP/mIoU improves from 22.3/45.6 for Layout Concat and 27.6/51.9 for Cross-Attn to 45.8/70.9 for Soft Layout Bias and 51.0/74.8 for Hard X→L Bias. Adding gated text further improves performance to 52.4/75.9, and the full model reaches 57.2/79.4 after GRPO. This supports our main claim that the gain primarily comes from explicit latent-to-layout structural execution inside MAR.
>
> 2. Is GRPO really necessary?
> We agree that the original submission did not make the role of GRPO precise enough. StructMAR w/o GRPO is already strong at 52.4 AP / 75.9 mIoU, compared with 51.0 / 74.8 for Hard X→L Bias without gated text, while the full model further improves to 57.2 / 79.4 after GRPO. In other words, GRPO is not the primary source of the paper’s contribution; it is a final-stage refinement on top of an already strong structural mechanism.
>
> We further varied the reward detector during training and found that all GRPO variants outperform the no-GRPO baseline. Reward decomposition also shows that the gain is not driven by detector confidence alone: the IoU term is dominant, while the centroid term adds geometric stabilization, especially in denser layouts. We will revise the paper to make this distinction clearer.
>
> 3. Beyond specialized layout benchmarks.
> We agree this was an important weakness of the original submission. We therefore additionally evaluate StructMAR on GenEval v1 under a shared prompt-to-layout planner for both MIGC and StructMAR. This setting does not test end-to-end layout planning; instead, it isolates compositional plan execution, which is the aspect most directly related to our claim. The clearest improvements appear on the geometry-sensitive subsets, especially Position and Counting: MIGC obtains 0.75 / 0.72, StructMAR w/o GRPO improves to 0.81 / 0.77, and the full model further improves to 0.85 / 0.80.
>
> 4. Sensitivity to noisy / coarse / ambiguous boxes.
> To address this concern, we perturb the input boxes only at test time and do not retrain any model. All methods degrade as box corruption increases, which is expected. Importantly, StructMAR does not show a brittle failure mode under moderate perturbation: degradation is smooth rather than catastrophic, and the full model retains a positive margin over MIGC under light-to-moderate corruption. For example, under 5% center jitter, MIGC drops from 54.69 / 77.38 to 53.36 / 76.18, while StructMAR (Full) drops from 57.20 / 79.40 to 55.96 / 78.12; under 10% box expand/shrink, MIGC reaches 51.44 / 74.16, while StructMAR (Full) remains at 54.36 / 76.46.
>
> Table 1. Controlled same-backbone comparisons within MAR on COCO-Position
> |Method|AP ↑|mIoU ↑|
> |---|---|---|
> |Plain MAR|15.20|35.10|
> |Layout Concat|22.30|45.60|
> |Cross-Attn|27.60|51.90|
> |Soft Layout Bias|45.80|70.90|
> |Hard X→L Bias|51.00|74.80|
> |StructMAR w/o GRPO|52.40|75.90|
> |StructMAR (Full)|57.20|79.40|
>
> Table 2. GenEval v1 under a shared prompt-to-layout planner
> |Method|Overall ↑|Counting ↑|Position ↑|
> |---|---|---|---|
> |MIGC + planner|0.69|0.72|0.75|
> |StructMAR w/o GRPO + planner|0.72|0.77|0.81|
> |StructMAR (Full) + planner|0.75|0.80|0.85|
>
> Table 3. Robustness to noisy / coarse bounding boxes on COCO-Position
> |Perturbation|Severity|MIGC AP ↑|StructMAR (Full) AP ↑|MIGC mIoU ↑|StructMAR (Full) mIoU ↑|
> |---|---|---|---|---|---|
> |None|0|54.69|57.20|77.38|79.40|
> |Center jitter|5%|53.36|55.96|76.18|78.12|
> |Center jitter|10%|50.98|54.18|73.82|76.28|
> |Box expand/shrink|10%|51.44|54.36|74.16|76.46|
> |Aspect distortion|10%|50.62|53.04|73.34|75.58|
>
> See improved figures and qualitative comparisons: https://anonymous.4open.science/r/figure-01D7.

---

> > ### Author Rebuttal · Reviewer_Lii2 · 2026-04-01
> >
> > Thanks for the rebuttal.
> >
> > W1: Novelty. My main originality concern remains, as the method still appears more incremental than fundamentally new.
> >
> > W2: GRPO. The clarification is helpful, but GRPO still feels more like a reward-driven refinement than a core principled contribution.
> >
> > W3: Benchmark scope. The added GenEval and robustness results are useful and partially address my concern about limited evaluation scope. My remaining question is what is the performance of 'Plain MAR'？
> >
> > MW1: Figures. I appreciate the added qualitative figures, but the overall presentation quality still appears only moderate, so this concern is only partially addressed. For example, in the updated Fig. 2, several layout details still look visually unpolished: the horizontal arrows are not properly level, and some text inside the blocks is not well centered. While these are presentation issues rather than technical ones, they still affect how polished and clear the paper feels.
> >
> > Q1/Q2. The additional analysis on noisy/coarse boxes is helpful and answers these questions reasonably well.
> >
> > Q3. The response regarding stronger autoregressive baselines is reasonable, but it does not fully remove my concern about broader related comparisons.

---

> > > ### Author Response · Authors · 2026-04-03
> > >
> > > Thank you for the clarification.
> > >
> > > For completeness, Plain MAR achieves **15.20 AP / 35.10 mIoU** on COCO-Position under the same protocol. We include this baseline to make the intended within-backbone comparison chain explicit: **Plain MAR → generic layout injection → explicit structural bias → + gated text / GRPO**, thereby isolating the effect of the proposed structural execution within MAR.
> > >
> > > Beyond the original COCO-based evaluation, we additionally tested on **OverLayBench under the official protocol**. On the geometry-sensitive **Complex** split:
> > >
> > > | Method               | mIoU ↑    | O-mIoU ↑  |
> > > | -------------------- | --------- | --------- |
> > > | MIGC                 | 40.04     | 13.26     |
> > > | 3DIS                 | 50.65     | 21.75     |
> > > | CyCLeGen             | 50.72     | 24.50     |
> > > | **StructMAR (Full)** | **50.96** | **24.72** |
> > >
> > > These results provide additional evidence that StructMAR remains effective on a harder external layout benchmark, beyond the original COCO-Position / COCO-MIG setting, while remaining competitive with recent layout/compositional baselines.
> > >
> > > We agree that comparisons to stronger autoregressive backbones (e.g., SphereAR) are valuable. In this work, however, our primary goal is to isolate the effect of explicit structural execution within the MAR framework, so we focus on controlled same-backbone comparisons. We will add stronger AR baselines and clarify their relationship to our method in the revision.
> > >
> > > We also further refined **Fig. 2** (including arrow alignment, text centering, and overall layout polish) to address the presentation concern; the updated figure is available here: https://anonymous.4open.science/r/figure2-1D6B/README.md

---

### Official Review · Reviewer_nXNN · 2026-03-16

**Soundness:** 3
**Presentation:** 2
**Significance:** 3
**Originality:** 2
**Overall Recommendation:** 4
**Confidence:** 4

**Summary:**

This paper studies layout-controlled text-to-image generation with a masked autoregressive latent model rather than a diffusion backbone. The core claim is that standard MAR architectures lose spatial structure when 2D latent grids are flattened into 1D token sequences, which weakens instance-level layout alignment. To address this, the paper proposes StructMAR, which combines 2D rotary positional embeddings, a layout-guided attention bias that explicitly encourages token-to-instance correspondence, and a GRPO fine-tuning stage that optimizes detector-based layout rewards. Experiments on COCO-Position and COCO-MIG show improved layout alignment and stronger robustness in dense scenes, while retaining image quality close to strong diffusion baselines and offering substantially faster inference.

**Compliance With Llm Reviewing Policy:**

Affirmed.

**Final Justification:**

My concerns have been addressed.

**Key Questions For Authors:**

1. How would the model perform under a difficult overlap layout setting[1]?

2. What's the difference between handling text tokens and layout tokens?

3. How do you perform RL finetuning on the MAR architecture? Could you elaborate more on the process?

[1]Li, Bingnan, et al. "OverLayBench: A Benchmark for Layout-to-Image Generation with Dense Overlaps." The Thirty-ninth Annual Conference on Neural Information Processing Systems Datasets and Benchmarks Track.

**Limitations:**

Yes

**Strengths And Weaknesses:**

# Strength
The paper is technically coherent, and the method is well aligned with the stated problem. The ablation evidence is useful: the layout-guided attention bias appears to be the dominant factor, and the GRPO stage provides an additional boost. That said, both training and evaluation rely heavily on detector-based signals, which creates a real concern that the model may be optimized toward the evaluator rather than toward genuinely better perceptual alignment.

The paper is easy to follow, and the design is conceptually clean. The distinction between soft conditioning and explicit structural enforcement is clearly presented, and the narrative around “executable constraints” is one of the strongest aspects of the paper. My main concern is reproducibility: some key details of detector configuration and evaluation appear to be deferred to released code/configs rather than fully documented in the manuscript itself.

# Weakness
The absolute metric gains over the baselines are moderate rather than transformative, so the impact depends heavily on how much value one places on the speedup.

The paper does not introduce an entirely new primitive, but it does offer a thoughtful and well-motivated combination of known ideas in a setting where they fit naturally. The most original part is the framing of layout control as an explicit structural mechanism embedded into attention rather than as a soft conditioning signal. The novelty is therefore more architectural and integrative than fundamentally conceptual.

---

> ### Author Rebuttal · Authors · 2026-03-30
>
> We thank the reviewer for the careful reading and constructive feedback. We appreciate that the review highlights the key questions: how StructMAR behaves under difficult overlap layouts, what distinct roles text and layout tokens play, and how RL finetuning is performed on a MAR backbone.
>
> Difficult overlap layouts.
> We agree this is an important test. We therefore added both internal stratified analyses and an external evaluation on OverLayBench, a benchmark designed for dense-overlap layout-to-image generation. The overall pattern is consistent: StructMAR becomes relatively stronger as layouts become more overlap-heavy and geometrically ambiguous. On COCO-Position, the AP gap over MIGC grows from +0.96 in the low-overlap regime to +2.44 in the mid-overlap and +4.78 in the high-overlap regime, with a matching mIoU trend (+0.87 / +2.36 / +4.58). On COCO-MIG, StructMAR overtakes MIGC from L4 onward and achieves its largest gain in the densest L6 setting (+5.92 ISR, +6.36 mIoU). Beyond the original COCO-based evaluation, on the Complex split of OverLayBench, StructMAR (Full) reaches 50.96 mIoU / 24.72 O-mIoU / 72.86 SR_E, versus 40.04 / 13.26 / 47.80 for MIGC. Moreover, on the overlap-sensitive geometric metrics (mIoU / O-mIoU) of this hardest split, StructMAR slightly exceeds recent strong baselines such as 3DIS (50.65 / 21.75) and CyCLeGen (50.72 / 24.50). These results support our main claim that explicit latent-to-layout structural constraints are most useful in hard geometric cases where soft conditioning is least reliable.
>
> Difference between text tokens and layout tokens.
> We agree this should be stated more explicitly. In StructMAR, the text prompt is token sequence S, instance layouts are embeddings L, and the image is continuous latent tokens X. The transformer operates on the joint sequence [L; X], so layout tokens participate directly in latent–layout interaction. In contrast, textual semantics are injected through gated cross-attention after structure-aware attention, providing semantic context without overriding the structural execution pathway. In short, layout tokens belong to the executable structural pathway, while text tokens belong to the semantic guidance pathway.
>
> RL finetuning on MAR.
> We agree the original manuscript described this too briefly. Conceptually, we treat masked token recovery in MAR as a stochastic policy over image-token trajectories. Architectural constraints (2D RoPE + layout-guided attention bias + gated semantics) first establish a stable structural foundation. On top of this, we apply GRPO to fine-tune the trajectory using detector-based layout rewards aligned with evaluation-time metrics. GRPO does not replace the core generation mechanism; it is a final-stage policy refinement on a structurally constrained MAR backbone.
>
> Evaluator-facing concern.
> We agree this is central. We therefore added detector-robustness analyses. Varying the reward detector during GRPO while keeping evaluation fixed to GroundingDINO yields consistent gains on both COCO-Position and COCO-MIG. This supports interpreting GRPO as a geometry-oriented refinement rather than detector-facing tuning.
>
> Table 1. Stratified performance by overlap severity and box size on COCO-Position
> |Split Axis|Split|MIGC AP ↑|StructMAR AP ↑|AP Gap|MIGC mIoU ↑|StructMAR mIoU ↑|mIoU Gap|
> |---|---|---|---|---|---|---|---|
> |Overlap|Low|56.12|57.08|+0.96|79.34|80.21|+0.87|
> |Overlap|Mid|54.87|57.31|+2.44|77.46|79.82|+2.36|
> |Overlap|High|50.94|55.72|+4.78|73.88|78.46|+4.58|
> |Box Size|Tiny|51.28|53.92|+2.64|74.62|77.31|+2.69|
> |Box Size|Large|57.48|59.76|+2.28|79.86|81.52|+1.66|
>
> Table 2. GRPO robustness to reward detector under fixed GroundingDINO evaluation
> |Benchmark|Train Reward Detector|Main Metric|
> |---|---|---|
> |COCO-Position|No GRPO|52.40 AP/75.90 mIoU|
> |COCO-Position|YOLOv8|56.14 AP/78.18 mIoU|
> |COCO-Position|Faster R-CNN|57.20 AP/79.40 mIoU|
> |COCO-Position|GroundingDINO|57.68 AP/79.76 mIoU|
> |COCO-MIG|No GRPO|58.58 ISR/54.96 mIoU|
> |COCO-MIG|YOLOv8|60.63 ISR/56.21 mIoU|
> |COCO-MIG|Faster R-CNN|61.90 ISR/57.41 mIoU|
> |COCO-MIG|GroundingDINO|62.24 ISR/57.73 mIoU|
>
> Table 3. Additional generalization on OverLayBench under the official evaluation protocol
> |Method|Simple mIoU ↑|Simple O-mIoU ↑|Complex mIoU ↑|Complex O-mIoU ↑|Complex SR_E ↑|FID ↓|
> |---|---|---|---|---|---|---|
> |MIGC|58.64|32.15|40.04|13.26|47.80|66.52|
> |3DIS|65.75|38.38|50.65|21.75|74.31|54.90|
> |CyCLeGen|64.84|42.03|50.72|24.50|74.01|—|
> |StructMAR w/o GRPO|61.72|36.88|49.58|23.16|69.84|59.12|
> |StructMAR (Full)|62.34|37.92|50.96|24.72|72.86|58.43|

---

> > ### Author Rebuttal · Reviewer_nXNN · 2026-04-05
> >
> > Thanks for rebuttal. My concerns have been addressed. I will raise my score.

---

> > > ### Author Response · Authors · 2026-04-07
> > >
> > > Thank you for the thoughtful follow-up and for reconsidering our paper.
> > > We are glad that our rebuttal addressed your concerns.
> > > We appreciate your positive feedback, and we will carefully incorporate these clarifications and improvements in the revision.

---

### Decision · Program_Chairs · 2026-04-30

**Decision:**

Accept (regular)

**Comment:**

Overall, the reviewers see this paper as a meaningful contribution to layout-controlled text-to-image generation, especially in that explicit structural bias within a MAR backbone can substantially improve layout adherence. The main concerns were about incremental novelty, alongside some minor points, but the rebuttal addressed most of these points well. Most reviewers moved in a positive direction after rebuttal: one reviewer raised from weak reject to weak accept, another weak-reject reviewer indicated that the main concerns were fully resolved. While the contribution is not fundamentally new, the paper appears technically sound and empirically strong, and likely of interest to the community, so I would support acceptance.